# Regional and temporal patterns of partisan polarization during the COVID-19 pandemic in the United States and Canada

Zachary Yang[1,3]*, Anne Imouza[2], Maximilian Puelma Touzel[3], Cécile Amadoro[3,4], Gabrielle Desrosiers-Brisebois[3], Kellin Pelrine[1,3], Sacha Levy[1,3], Jean-François Godbout[3,4], Reihaneh Rabbany[1,3]

1 Department of Computer Science, McGill University, Montreal, Quebec, Canada, 2 Department of Political Science, McGill University, Montreal, Quebec, Canada, 3 Montreal Institute for Learning Algorithms, Montreal, Quebec, Canada, 4 Department of Political Science, University of Montreal, Montreal, Quebec, Canada

* zachary.yang@mail.mcgill.ca

## Abstract

During the COVID-19 pandemic, public health measures emerged as highly polarizing topics in online discourse, with debates intensifying around specific policy implementations and events. This study introduces a novel computational approach to measure subnational and event-driven variation in partisan polarization and explores these dynamics across the United States and Canada. Analyzing over 50 million tweets from late 2020—a critical period of polarizing discourse during the pandemic's early phase—we examine regional variations in discussions surrounding three key health interventions: lockdowns, masks, and vaccines. Our analysis reveals that politically conservative regions exhibited significantly higher levels of partisan polarization in both countries, with this effect particularly pronounced in the United States. We demonstrate a strong negative correlation between regional vaccination rates and the degree of polarization in vaccine-related discussions in the U.S., suggesting tangible public health implications of online partisan division. This relationship was notably weaker in Canada, pointing to important cross-national differences in how political polarization manifests and impacts health behaviors. By tracking the temporal evolution of polarization, we identify distinct spikes linked to specific political events and policy announcements. These polarization surges typically lasted only a few days, revealing the dynamic nature of online partisan discourse. The geographic heterogeneity in polarization patterns—with certain conservative states showing unexpectedly low polarization and some liberal states displaying high polarization—highlights the complex interplay between political ideology, policy implementation, and public response during health crises. While our polarization index reflects the discourse of politically engaged Twitter users, it nevertheless captures key dynamics of online debate that influence public narratives and align with observable regional outcomes. Our findings suggest that online discussions both reflect and potentially drive rapid

**Data availability statement:** Trained language models will also be hosted publicly on HuggingFace. Analysis code, preprocessing scripts, and list of tweet ids & profile ids will be released on GitHub. The full twitter dataset cannot be shared publicly because of Twitter's policy.

**Funding:** This research is supported by CIFAR AI Catalyst Grants and Canada CIFAR AI Research Chair funding.

**Competing interests:** The authors have declared that no competing interests exist.

changes in public opinion, with measurable consequences for regional public health outcomes. This computational framework for quantifying polarization provides a valuable tool for researchers and policymakers to understand, monitor, and potentially address partisan divisions during public health emergencies and beyond.

## Introduction

Partisan polarization has become an increasingly prominent feature of democracies worldwide [1]. In the United States, the divide between Democrats and Republicans has steadily intensified over decades [2,3], reaching unprecedented levels during the 2020 presidential election [1,4]. This polarization fundamentally shaped how individuals responded to the COVID-19 pandemic, influencing their assessment of viral threats and their compliance with public health measures [5–7].

Extensive research now confirms that in the U.S., Democratic Party supporters were consistently more likely to adhere to social distancing guidelines [8–10], wear masks [11,12], and get vaccinated [13–15] compared to their Republican counterparts. This partisan divergence in pandemic response is not unique to the United States [16,17]. Similar patterns have emerged in other countries, including Canada, where Liberal Party supporters demonstrated higher compliance with COVID-19 guidelines than supporters of the Conservative Party or the populist People's Party [18,19].

While national governments in both the United States and Canada attempted to coordinate pandemic responses, subnational jurisdictions implemented remarkably diverse public health interventions. Canadian provinces adopted a spectrum of policies ranging from comprehensive lockdowns and school closures to narrowly targeted measures focused on specific populations [20]. Similarly in the United States, state-level responses varied dramatically—some states implemented strict lockdown orders and mask mandates, while others minimized restrictions on social activities [21]. These regional policy differences quickly became politicized along partisan lines [17], with social media platforms serving as the primary arena where this politicization unfolded [10,22–24].

A growing body of research confirms that online pandemic discussions [9,25–27] exhibited distinct regional patterns characterized by the same partisan animosity that influenced heterogeneous implementation of public health measures [25] and subsequent epidemiological outcomes [9]. These partisan divisions contributed to accelerating polarization on social media platforms [24,25,27]. The intense reactions from both supporters and opponents of public health measures [28] suggest that public opinion was significantly shaped by local political dynamics and the geographic progression of the pandemic [29]. Political discussions also vary naturally across socio-demographic groups, with each group developing its own lexicon and issue priorities [30–32]. This variation is well-documented not only among political elites but also in public discourse across the U.S. and Canada, where demographic and regional cleavages strongly shape patterns of partisan communication [33].

This regional heterogeneity necessitates studying polarization through a multidimensional lens that accounts for specific events, topics, and geographic variation.

Our work addresses this need by providing a comprehensive, large-scale study of geographic and event-driven variation in online partisan polarization surrounding COVID-19 discussions across American states and Canadian provinces. We introduce a methodological framework to reliably measure polarization in public discourse at fine-grained resolution, leveraging the vast quantity of human-generated text available on digital platforms. Specifically, we analyze millions of tweets to determine how polarization correlates with: (1) the ideological leanings of different regions; (2) exposure to conspiracy theory-related content; and (3) vaccination rates. While our study focuses on the COVID-19 context, the proposed methodology for quantifying partisan polarization at scale can be adapted to examine large-scale discussions across various temporal, spatial, and topical dimensions.

Recent computational approaches increasingly draw on text embedding methods to study polarization, using contextualized representations to detect partisan divides across topics and communities [34–36]. Embedding-based models capture latent semantic distinctions between ideological groups, providing methodological advantages that complement traditional lexicon- or dictionary-based strategies. At the same time, network-based approaches such as inferring ideology from follower or retweet structures have proven highly effective in prior work [37,38,39]. However, access to user- and network-level data has become increasingly restricted, particularly under recent platform policy changes. Consequently, our study adopts a strictly text-based method, which remains both accessible and scalable for analyzing large-scale digital discourse.

The paper is organized as follows. First, we outline our approach to extracting region- and time-resolved polarization data from Twitter (X). In this section, we describe our method for classifying users as conservative or liberal and justify our choice of topic-conditioned language dissimilarity as a proxy for partisan polarization. Next, we present results from applying our approach to a large-scale dataset collected in 2020, filtered through three prominent pandemic-related public health interventions: lockdowns, masks, and vaccines. Our findings reveal that conservative regions in both countries exhibited higher overall polarization levels regarding these topics. We also identify a strong negative correlation between regional vaccination rates in the U.S. and polarization levels in online vaccine discussions. We conclude with a discussion of methodological limitations and promising new applications of our approach.

## Methods

We developed a novel method to measure geographically-resolved partisan polarization over time using large-scale social media message datasets (see Fig 1). Our approach addresses a fundamental challenge in measuring

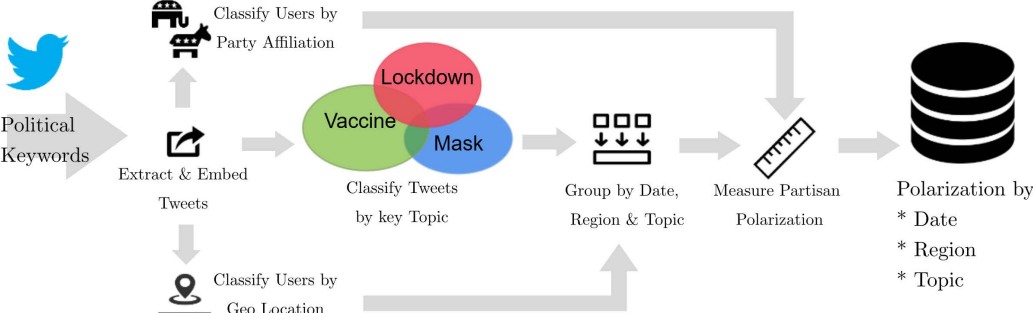

**Fig 1. Overview of estimating partisan polarization over date, region and topic.** Using a predefined set of political keywords, we extract and embed tweets, geolocate users, and classify them by party affiliation. Tweets are then categorized by key topic (lockdowns, masks, and vaccines) allowing us to measure partisan polarization over time (daily) and across regions (weekly) in both Canada and the U.S.

polarization: the language of political discussions across socio-demographic groups naturally varies, with each group having their own lexicon [40]. Therefore, dissimilar language alone does not necessarily indicate polarized positions.

However, when examining discussions on the same contentious topic, language dissimilarity between different partisan groups more likely reflects alternative semantic understandings of that specific topic—a phenomenon we hypothesize correlates strongly with polarization. While this correlation may be weakened by linguistic differences not fully captured by any particular semantic model, we expect that more expressive language models provide stronger indicators of polarization.

Our methodology leverages the demonstrated capacity of modern vector embeddings to represent language semantics. Specifically, we transform social media posts using RoBERTa, an open-source transformer-based language model [41]. We then measure polarization by quantifying how far apart the language of left and right-leaning users is positioned in this embedding space, using the C-index [42], a robust clustering measure. This index calculates the average pairwise distances between embedded partisan users within a partisan group relative to the average of the largest and smallest distances overall.

To identify partisan users, we developed and validated a two-step approach for label propagation that classifies users as conservative or liberal by cross-indexing multiple metadata sources. We also developed a method to geolocate users to capture polarization's geographic heterogeneity.

### Application to late 2020 pandemic discourse in the United States & Canada

We collected a large-scale dataset of COVID-19 political discussions on Twitter (now X) occurring between October 9, 2020, to January 4, 2021, comprising approximately 12.5 million tweets linked to the United States and 46.6 million tweets linked to Canada.

Users were geolocated based on their provided location information and classified by their declared party affiliation. For the United States, we identified supporters of the Democratic (left) and Republican (right) parties. For Canada, we grouped users into left (Liberal Party, New Democratic Party, and Green Party) and right (Conservative Party and People's Party) partisan families.

We verified that our data provided a politically balanced representation of users across different regions in both countries by comparing with official population census data and election results. For each user, we computed a vector representation of their language used in social media messages during this period. We then calculated our polarization metric conditioning on region, time, and topic.

Our analysis focused on three specific pandemic-related topics: lockdowns, masks, and vaccines. These topics were selected for their salience in polarized discourse [27,43,44], and because they represent different types of interventions (group behavior, individual behavior, and medicine, respectively). Based on these measurements, we compared the polarization observed in different American states and Canadian provinces over time for each topic, and examined correlations with epidemiological data and conspiracy-related content.

### Data collection

We collected Twitter (X) data, COVID-19 case data from official government websites, vaccination rates from public health records, and U.S. voter registration records from state government portals. All data were obtained and analyzed in compliance with the respective platforms' and websites' terms of service and applicable laws.

**2.2.1. Twitter (X) data.** We used Twitter's (X) official API to collect 1% of real-time tweets for the United States and Canada from October 9, 2020 to January 4, 2021. This dataset comprised 387,090,097 tweets from 23,758,112 users for the United States and 231,841,790 tweets from 4,765,115 users for Canada. We filtered the data using political keywords:

**United States**: 'JoeBiden', 'DonaldTrump', 'Biden', 'Trump', 'vote', 'election', '2020Elections', 'Elections2020', 'PresidentElectJoe', 'MAGA', 'BidenHaris2020', 'Election2020'.

**Canada**: 'trudeau', 'legault', 'doug ford', 'pallister', 'horgan', 'scott moe', 'jason kenney', 'dwight ball', 'blaine higgs', 'stephan mcneil', 'cdnpoli', 'canpol', 'cdnmedia', 'mcga', 'covidcanada' and all combinations of 'covid' or 'coronavirus' as the prefix and the (full & abbreviated) name of each provinces and territories as the suffix.

**2.2.2. COVID-19 epidemic data.** We used officially reported COVID-19 new cases and death data from each country. For the United States, we sourced data from the Centers for Disease Control and Prevention (CDC). For Canada, we use the data reported by the Canadian government.

**2.2.3. COVID-19 vaccination rate.** We used officially reported vaccination rates from one year after our Twitter data collection period, as COVID-19 vaccines were only created and approved at the very end of our initial data collection. The vaccination rates represent those who obtained at least two doses. For the United States, we use the *people_fully_vaccinated_per_hundred* reported in the COVID Data Tracker from the CDC. For Canada, this is the *numtotal_fully* from the government's vaccine coverage map. We normalize this column by Canada's 2021 population per province or territory.

**Classifying tweets by topics**

We focused on three key COVID-19 topics—lockdowns, masks, and vaccines—plus conspiracy theories. For each topic, tweets were classified as *relevant* or *irrelevant* based on whether they contained at least one topic-specific keyword. For conspiracy-related tweets, *relevant* indicates that the content pertains to COVID-19 conspiracy theories, including tweets that either promote the conspiracy or debunk / refute it, i.e., any tweet discussing the theory in either context. A tweet could belong to multiple topics.

Our classification approach followed these steps:

1. We extracted all hashtags within our dataset, ordered by frequency, and discarded those appearing fewer than 100 times (yielding 18,000 hashtags for the United States and 3,600 for Canada).

2. Two political scientists manually annotated this list with topic and relevance labels, narrowing it to 631 relevant hashtags.

3. We merged these with hashtags identified in previous studies for the same topics [45–47].

This process yielded 12,552,213 tweets from 2,657,355 users for the United States and 46,636,206 tweets from 1,757,675 users for Canada that shared COVID-19-related content.

We then domain-adapted RoBERTa-base models [41] on the COVID-19 tweets from each country's dataset through self-supervised learning (predicting masked words within tweets), creating country-specific pre-trained language models.

For topic classification validation, we randomly sampled 200 *relevant* and 200 *irrelevant* tweets per topic from each dataset (1,600 tweets total). Two political scientists independently reviewed each tweet to determine relevance, discarding tweets where consensus couldn't be reached. We then fine-tuned the respective domain-adapted RoBERTa models to classify by topics. We split the labeled data into 60%−20%−20% train–validation–test sets. During training, we also incorporated hashtag-annotated tweets as weak supervision, assigning them a reduced loss weight of 0.9 compared to 1.0 for gold-standard labels.

Table 1 reports the performance of the fine-tuned RoBERTa classifiers against the manually annotated validation set (columns 2–4), showing near-perfect F1-scores across topics and high inter-annotator agreement (Cohen's Kappa). The final column shows, after training, the total number of tweets each classifier predicted as relevant to that topic in the full dataset.

**Table 1. Tweet topic classification metrics.**

**United States**

| Topic | Relevant | Irrelevant | Cohen | F1-Score | # of Tweets |
|---|---|---|---|---|---|
| Lockdown | 126 | 199 | 0.63 | 100.00 ± 0.00 | 897,565 |
| Mask | 197 | 200 | 0.98 | 99.49 ± 0.62 | 1,562,706 |
| Vaccine | 192 | 201 | 0.96 | 100.00 ± 0.00 | 1,541,360 |
| Conspiracy | 195 | 155 | 0.75 | 95.55 ± 2.39 | 926,389 |

Canada

| Topic | Relevant | Irrelevant | Cohen | F1-Score | # of Tweets |
|---|---|---|---|---|---|
| Lockdown | 170 | 356 | 0.73 | 97.13 ± 1.57 | 1,553,984 |
| Mask | 292 | 282 | 0.91 | 98.48 ± 1.24 | 1,994,293 |
| Vaccine | 326 | 248 | 0.91 | 99.84 ± 0.31 | 2,145,549 |
| Conspiracy | 338 | 171 | 0.67 | 97.20 ± 0.71 | 16,575,934 |

For evaluation, we used the same set of 200 relevant and 200 irrelevant manually annotated tweets per topic. F1-scores are reported over five training runs with different random seed initializations, using identical training data in each run.

## Classifying users by geo-location

To ensure balanced regional representation in our data, we geolocated users with explicit location information in their profiles. We processed this free-form text field using Open Street Map and the ArcGIS API, which both return latitude and longitude coordinates when a location is found. We established a clear geolocation when both APIs returned coordinates within one degree of each other, finding this approach more accurate than using pre-trained Named Entity Recognition algorithms.

In total, we successfully geolocated **757,601** U.S. users with a strong correlation of **0.98** (n = 52, p = 9.27e-35, CI=[0.96, 0.99]) to official population census data, and **282,454** Canadian users with a strong correlation of **0.92** (n = 13, p = 6.20e-06, CI=[0.76, 0.98]). These high correlations confirm that each region is well-represented in our data (see Table 2).

## Classifying users by party affiliation

Our approach to determining users' party affiliation followed a two-step process, achieving a macro-F1-score of 91% for both countries (Table 3 and 4). For both classifiers, results are averaged over five random seeds, and we report the mean F1 score and its standard deviation.

1. **Profile classifier**: We first identified politically explicit users based on their profile descriptions, using partisan keywords for each country's major parties.

2. **Activity classifier**: We then used these profile-based classifications as labels to train a classifier based on user activity patterns.

**2.5.1. Profile classifier.** As a preprocessing step, we filter out users that are not politically explicit. *Politically explicit users* are those whose profile description contains at least one political keyword defined for any political party. For Canada, we focused on the five main political parties: Conservative, Green, Liberal, New Democratic Party and People's Party. For the United States, we focused on the Democratic and Republican parties. The following is the set of keywords we have per party:

**United States**:

*Democrat* - 'liberal,' 'progressive,' 'democrat,' 'biden'
*Republican* - 'conservative,' 'gop,' 'republican,' 'trump,'

**Table 2. Geolocated users number and correlation.**

**United States**

| Users | Total | Correlation |
|---|---|---|
| Geolocated | 757,601 | **0.98** (n = 52, p = 9.27e-35, CI=[0.96, 0.99]) |
| w/ Party Affiliation | 242,056 | **0.97** (n = 52, p = 2.51e-34, CI=[0.96, 0.99]) |

Canada

| Users | Total | Correlation |
|---|---|---|
| Geolocated | 282,454 | **0.92** (n = 13, p = 6.20e-06, CI=[0.76, 0.98]) |
| w/ Party Affiliation | 195,456 | **0.92** (n = 13, p = 6.83e-06, CI=[0.76, 0.98]) |

Correlation is done with the official 2021 population census for each country.

**Table 3. American user party affiliation classification.**

| | Party | Support | F1-Score | # of Users |
|---|---|---|---|---|
| Profile | Republican | 854 | 97.21 ± 0.66 | 86,989 |
| | Democrat | 928 | 97.40 ± 0.63 | 82,923 |
| Activity | Republican | 10,583 | 92.98 ± 0.18 | 239,449 |
| | Democrat | 10,226 | 92.99 ± 0.16 | 145,733 |
| Combined | Republican | – | – | 336,231 |
| | Democrat | – | – | 426,933 |

Cohen Kappa score of 0.76. We report the mean F1 score and its standard deviation over 5 runs.

**Table 4. Canadian user party affiliation classification.**

| | Party | Support | F1-Score | # of Users |
|---|---|---|---|---|
| Profile | CPC | 98 | 92.93 ± 1.12 | 1,769 |
| | GPC | 60 | 88.50 ± 1.54 | 97 |
| | LPC | 100 | 90.89 ± 1.34 | 783 |
| | NDP | 124 | 93.44 ± 0.53 | 370 |
| | NO_PARTY | 105 | 86.16 ± 1.97 | 667 |
| | PPC | 95 | 94.09 ± 1.34 | 402 |
| Activity | RPF | 628 | 93.85 ± 0.47 | 193,225 |
| | LPF | 357 | 89.10 ± 0.73 | 299,836 |
| Combined | RPF | – | – | 196,338 |
| | LPF | – | – | 302,023 |

Cohen Kappa score of 0.74. We report the mean F1 score and its standard deviation over 5 runs.

**Canada:**

   *Conservative* - 'erin o'toole', 'andrew scheer', 'conservative', 'conservative party', 'cpc', 'cpc2021', 'cpc2019', 'conservative party of canada'
   *Green* - 'annamie paul', 'green party', 'gpc', 'gpc2019', 'gpc2021', 'green party of canada'
   *Liberal* - 'justin trudeau', 'liberal', 'liberal party', 'lpc', 'lpc2021', 'lpc2021', 'lpc2019', 'liberal party of canada'

*New Democratic Party* - 'jagmeeet singh', 'new democrat', 'new democrats', 'new democratic party', 'ndp', 'ndp2021', 'ndp2019'

*People's Party* - 'maxime bernier', 'people's party', 'ppc', 'ppc2019', 'ppc2021', 'people's party of canada'

We randomly selected *politically explicit* users for manual annotation by two political scientists, achieving Cohen's Kappa scores of 0.74 for Canada and 0.76 for the United States. Using an 80−20 train-test split, we then trained a RoBERTa-large model to determine user party affiliation based on profile text.

**2.5.2. Activity classifier.** For the activity classifier, we generated user embeddings by aggregating tweet embeddings from each user (using mean pooling), then trained a multi-layer perceptron to predict party affiliation. We filtered users based on their activity level (number of COVID-related tweets), with optimal thresholds determined to be 5 tweets for Canada and 10 tweets for the United States. We evaluated thresholds of 1, 3, 5, 10, and 20 tweets, selecting the value that balanced high F1-score with sufficient user coverage. Thresholds beyond this point offered minimal F1 improvement but excluded a large number of users, so we opted for the more balanced choice.

Specifically for Canada, we found that the MLP could not distinguish the parties sufficiently – the F1-score was not satisfactory as parties within the liberal (left) party family and conservative (right) party family was easily confused as shown in the confusion matrix in Table 5. Hence, we grouped the parties based on their partisan leaning. The liberal (left) party family included the Liberal Party, New Democratic Party and Green Party while the conservative (right) party family included the Conservative Party and People's Party. We show the confusion matrix for after the merge in Table 6. We also removed supporters of other minor parties and the Bloc Quebecois.

**2.5.3. External validation with election results.** We further validated our party affiliation classifications by comparing them with real-world election data. For the U.S. and Canada, we calculated the correlation between the ratio of liberal to conservative-labeled users per region in our dataset compared to official election results, obtaining strong correlations of **0.802** for the United States and **0.815** for Canada (visualized in Fig 3 and Fig 2).

**2.5.4. External validation with voter registration records.** For further external validation of our U.S. party affiliation predictions, we matched Twitter users with primary voter registration records available for five states (Ohio, New York, Florida, Arkansas, North Carolina) and Washington DC.

Using our geolocation and voter record matching, we successfully matched over 30,000 users across these regions with their voter records (Table 7). Specifically, we obtain the party affiliation of unique users in each state by md5-hashing

**Table 5. Canadian user party affiliation activity classifier confusion matrix.**

|     | CPC | PPC | GPC | LPC | NDP |
|-----|-----|-----|-----|-----|-----|
| CPC | 398 | 24 | 1 | 29 | 6 |
| PPC | 59 | 49 | 0 | 1 | 1 |
| GPC | 3 | 0 | 6 | 10 | 6 |
| LPC | 18 | 3 | 2 | 161 | 26 |
| NDP | 5 | 1 | 1 | 19 | 58 |

**Table 6. Canadian user party affiliation activity classifier confusion matrix.**

|     | Right Party Family | Left Party Family |
|-----|-----|-----|
| Right Party Family | 530 | 38 |
| Left Party Family | 27 | 289 |

Liberal (left) and Conservative (right) Party Family

 

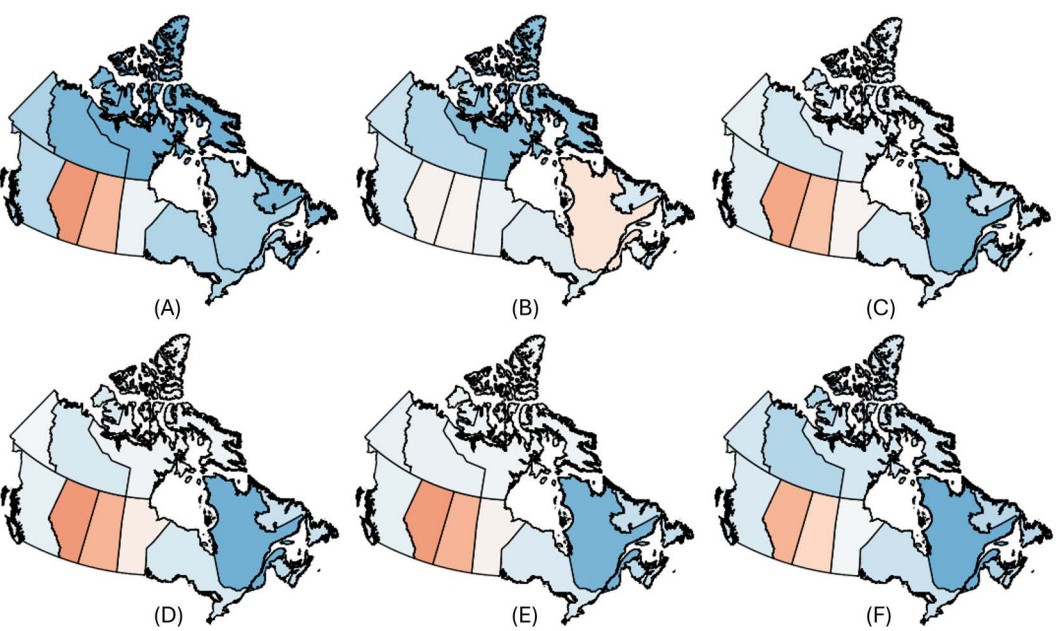

**Fig 2. Normalized distribution of the inferred user party affiliations compared to the Canadian 2021 election results.** A: Canadian 2021 election results. B: Geolocated users. C: Lockdown. D: Mask. E: Vaccine. F: Conspiracy.

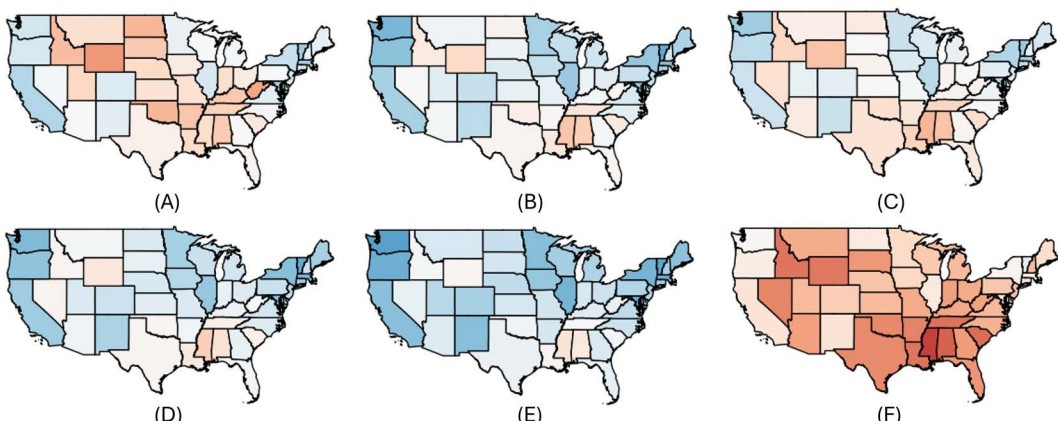

**Fig 3. Empirical distribution of the inferred user party affiliations compared to the US 2020 election results.** Interestingly, lockdown matches closer to the election results, mask and vaccine has a higher liberal ratio and conspiracy has a higher conservative ratio. A: Canadian 2021 election results. B: Geolocated users. C: Lockdown. D: Mask. E: Vaccine. F: Conspiracy.

their names and county to construct a key identifier. From our set of geolocated Twitter (X) users, we kept everyone that belonged to one of the five states or DC, and we removed those whose location could not be retrieved. Finally, we matched the most recent voter party affiliation records from the registration data to the unique Twitter (X) users that matched both the county and either the first name and last name or the first, middle and last name. We pre-processed the user's name on Twitter (X) to remove emojis. After matching, we removed users not affiliated with either of the two major parties and users whose name matched with more than one record per county (indicating a non-unique match).

**Table 7. Users matched to their voter registration.**

| State | Voters* | Users* | Matched | Democrat | Republican | Other |
|---|---|---|---|---|---|---|
| Ohio | 7,771,590 | 4,913 | 1,431 | 320 | 193 | 917 |
| New York | 17,718,437 | 30,927 | 8,255 | 4,843 | 1,631 | 1,781 |
| Florida | 14,477,882 | 50,541 | 12,905 | 5,585 | 4,508 | 2,810 |
| Arkansas | 1,722,465 | 4,311 | 1,280 | 145 | 140 | 995 |
| District of Columbia | 510,026 | 17,661 | 2,538 | 1,929 | 153 | 456 |
| North Carolina | 8,004,814 | 20,761 | 6,050 | 2,450 | 1,655 | 1,945 |
| Total | | | 32,456 | 15,272 | 8,280 | 8,904 |

Voters* and Users* respectively correspond to the number of unique voters in the records and unique Twitter (X) users in our data.

Our profile classifier achieved 74.35% accuracy and our activity classifier achieved 73.35% accuracy when compared to these official records. While these accuracies are lower than our training model performance, they are in line with standards for this type of evaluation [37], especially considering that voter registrations may be outdated or not fully reflect current partisan leanings expressed online. We note that name-based matching can miss users due to nicknames, misspellings, or common names; that self-reported Twitter locations may be incomplete, imprecise, or even intentionally false (including the rare influence of VPNs); and that automated accounts (bots) can provide misleading location details and generate highly partisan content, which may artificially inflate observed partisan divides.

## Measuring partisan polarization

Given a set of political parties $\mathcal{P}$ and user embeddings $\mathcal{U} = \{u^{(1)}, u^{(2)}, \ldots, u^{(n)}\}$ where $u^{(i)} \in \mathbb{R}^{768}$ and $n$ is the number of users, we measured polarization as follows.

We first embed each tweet using our language model. To obtain a user embedding for a given topic, we aggregate all of the user's relevant tweet embeddings by averaging. This results in a single vector representation per user that captures the content they share about that topic

We then treat the set of user embeddings as clustered by political party and compute a polarization score based on cluster separation. Specifically, we first calculate the sum of inter-cluster distances (i.e., the distance and dispersion between each party):

$$S_w = \frac{1}{2} \sum_{p \in \mathcal{P}} \sum_{u,v \in p} \|u - v\|$$

(1)

We then normalize this value based on its minimum and maximum possible ranges, $S_{min}$ and $S_{max}$, which correspond to the sum of the $m$ smallest (resp. largest) distances between points in $\mathcal{U}$; where $m = \sum_{p \in \mathcal{P}} \frac{|p|(|p|-1)}{2}$.

Based on these, we define our polarization index *poli* as:

$$poli = \frac{S_{max} - S_w}{S_{max} - S_{min}}$$

(2)

This index ranges from 0 (no polarization) to 1 (extreme polarization where partisan groups are completely isolated from each other in the embedding space). Because our measure aggregates tweets into user-level embeddings, it captures overall partisan alignment at the user level, rather than fine-grained nuances in individual tweets. As a result, subtle rhetorical strategies such as sarcasm or indirect disagreement may be less apparent, while consistent linguistic cues, including negation, are preserved in the aggregated representation.

### 2.6.1. Approximation algorithm for computational efficiency.

Computing the exact polarization is computationally intensive ($O(n^2 \log(n^2))$), making it impractical for large datasets. We developed an approximation algorithm that sub-samples users while maintaining accuracy.

Our algorithm (Algorithm 1) repeatedly samples a fraction of users, calculates the polarization index on this sample, and increases the sample size until the coefficient of variation falls below a specified threshold (e.g., 10 [48]). One loop of the approximation has a time complexity of $O(rf^2 * n^2 \log((n)^2))$ where $r$ is the repeat count and $f$ is the fraction (e.g., 0.01). From our testing, we know that at large values of $n$, the fraction needed rarely increases, so only one loop is required. Therefore, this approach significantly reduces computation time ($rf^2$ for large values of $n$.) while maintaining high accuracy.

### Algorithm 1 Approximating *poli*

```
Require: 𝒰, fraction, epsilon, step_size, repeats
1: while cv_poli > epsilon do
2:    poli_indices ← []
3:    num_of_runs ← 0
4:    for i in repeats do
5:      𝒰ₛ ← sample(𝒰, fraction)
6:      poli_indices.append(poli(𝒰ₛ))
7:    end for
8:    mean_poli ← mean(poli_indices)
9:    std_poli ← std(poli_indices)
10:   cv_poli ← std_poli/mean_poli
11:   fraction ← fraction + step_size
12: end while
13: return mean_poli
```

We validated our approximation method on daily lockdown and vaccine tweets. As shown in Fig 4, the absolute error drops dramatically at around 3,000 users and falls below 0.001 at approximately 10,000 users. The time saved increases exponentially with the number of users, with the approximation rarely needing to increase the sampling fraction beyond 50,000 users. All results are averaged over 10 independent runs, and we report the mean and standard deviation for both absolute error and time saved.

These findings confirm that we can accurately approximate polarization for large-scale data that would be impossible to measure exactly due to memory constraints. Our sampling approach saves both time and memory exponentially as dataset size increases.

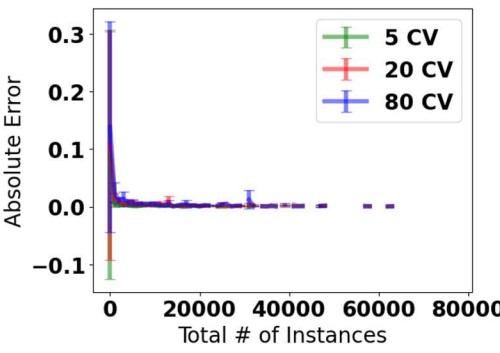 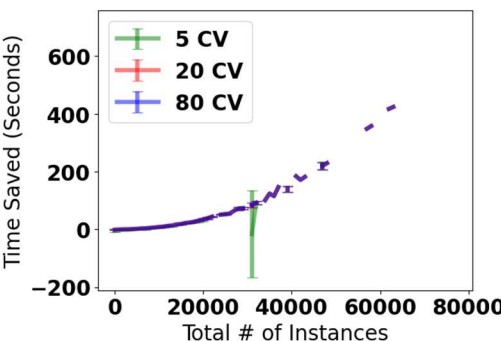

**Fig 4. Approximation of *poli*: Error and Runtime Tradeoff.** A: Approximation quality. For large enough data, the approximation error is close to 0.001.B: Time saved in seconds. Our approximation of polarization also exponentially saves time as the number of instances grows. For both panels, we plot the mean and standard deviation over 10 independent runs across different minimum coefficients of variation (CV).

We also explored the impact of changing the minimum coefficient of variation (CV). We start with a minimum sample size of 1% or *fraction* = 0.01. We keep the step_size constant at 0.01. For the first experiment, *repeats* is set to 10. For the second experiment, shown here in Fig 5 and Fig 6, *epsilon* is set to 0.05.

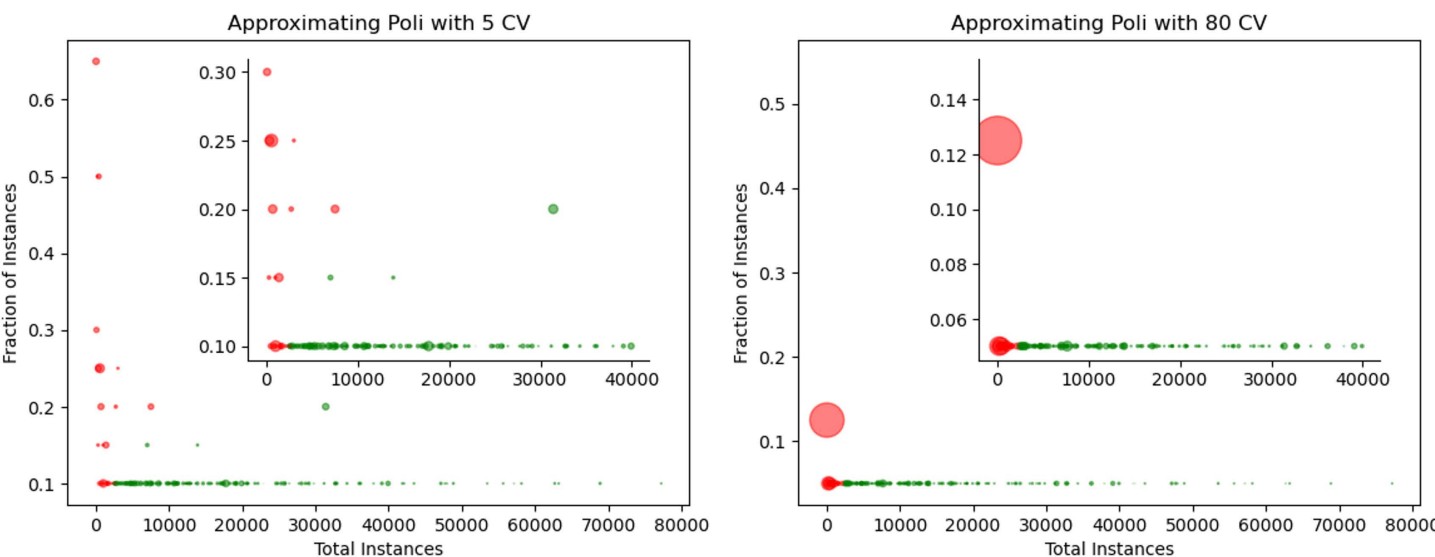

**Fig 5. Approximation of *poli*: Effect of Minimum Coefficient of Variation.** Green dots mean time was saved while red means time lost. The circle size is the absolute error times 1000. The inner plot zooms in and doubles the circle sizes for clarity. We observe that the fraction of instances needed increase as we decrease the needed minimum coefficient of variation. This follows our intuition as a larger sample of data is much easier to approximate the full data. We also that at lower minimum coefficient of variation, the less error we have (circles are much smaller). On all plots, we also see that we always save time (green circles) if there are more than 10,000 instances.

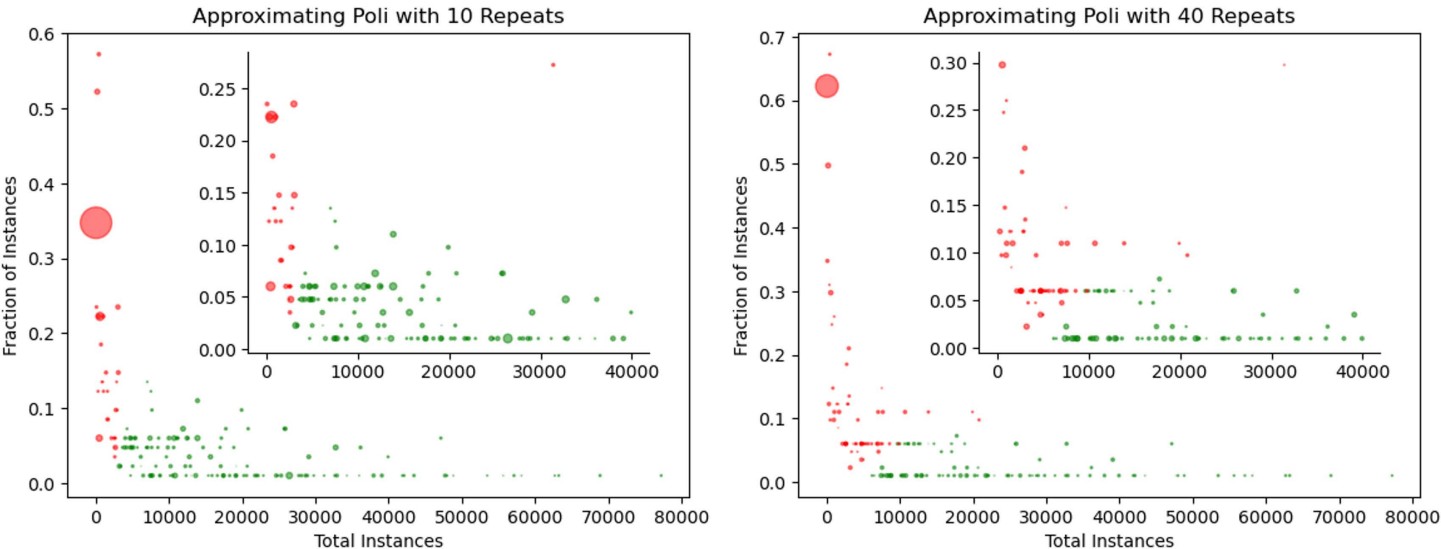

**Fig 6. Approximation of *poli*: Effect of Repeat Count.** As in the preceding figure, we observe that as we increase the number of repeats, the less error we have in our approximation (smaller circles). Likewise the amount of time saved decreases as we increase the repeat count (amount of red circles increase as repeats increase). We also observe a general slight increase in the fraction of instances needed as we increase the repeats.

## Results

We present our findings in a structured progression. First, we analyze geographical patterns of partisan polarization in the United States and Canada, demonstrating that conservative states and provinces exhibit higher polarization in online COVID-19 discourse. Next, we explore the relationship between vaccine-related polarization and actual vaccination rates across US states. We then examine temporal patterns of polarization at the national level in both countries, connecting peaks in polarization to specific events and analyzing correlations with vaccination data and conspiracy-related content. Finally, we investigate the relationship between conspiracy discourse and polarization.

### Regional variation in partisan polarization

Our geographical analysis reveals substantial regional heterogeneity in polarization levels across different COVID-19 topics (Fig 7 and Fig 8). This heterogeneity extends to conspiracy-related tweets (Fig 7D and Fig 8D).

When examined against partisan voting patterns (using the 2020 U.S. presidential election and 2019 Canadian federal election), we observe a clear trend: **conservative states and provinces show higher levels of polarization compared to their liberal counterparts**. This pattern is visible in the polarization rankings for American states and Canadian provinces (Fig 9 and Fig 10), with rankings applied separately to each topic (lockdowns, masks, vaccines) and aggregated as an "overall" score. Each region is associated with a color graded from blue to red based on the vote margin for the Republican party (US) or the conservative party family (Canada) obtained from the votes reported in the most recent election in their corresponding country. In these figures, a blue to red color gradient for conservative to progressive is used such that

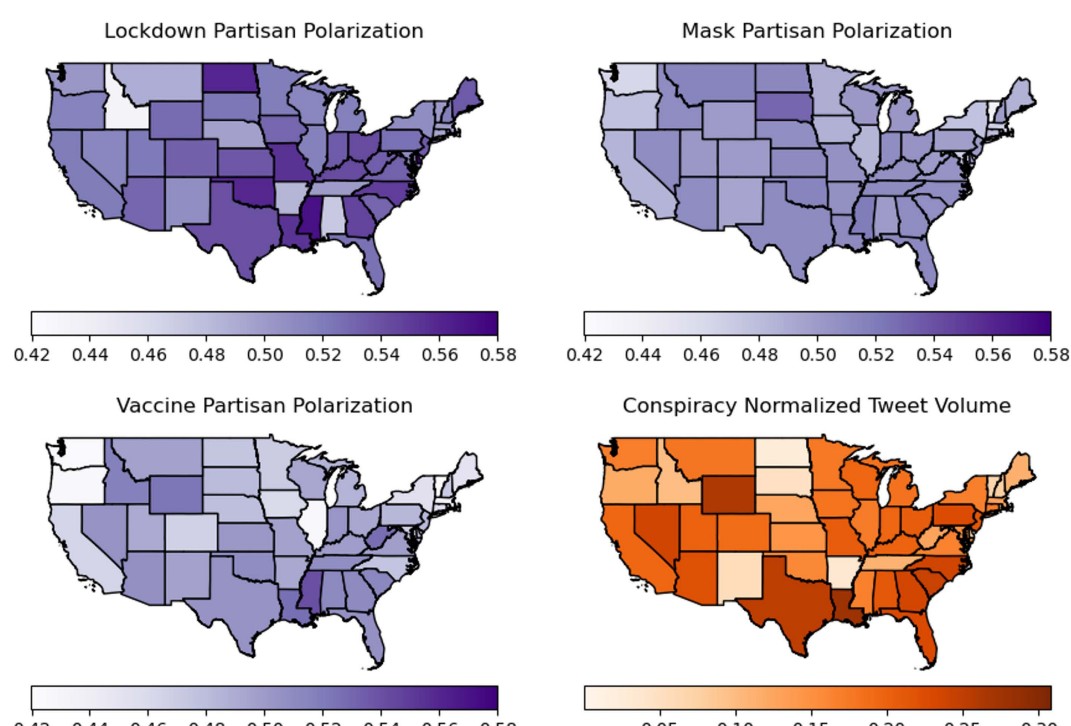

**Fig 7. Regional distribution of partisan polarization in the United States.** A: Lockdown. B: Mask. C: Vaccine. D: % of conspiracy-related tweets. Color intensity from light to dark gives the amount of polarization measured weekly between October 11, 2020 to January 3, 2021 and then averaged over the 12 weeks. We also report the average weekly percentage of conspiracy-related tweets that are posted from users in each region in panel (D).

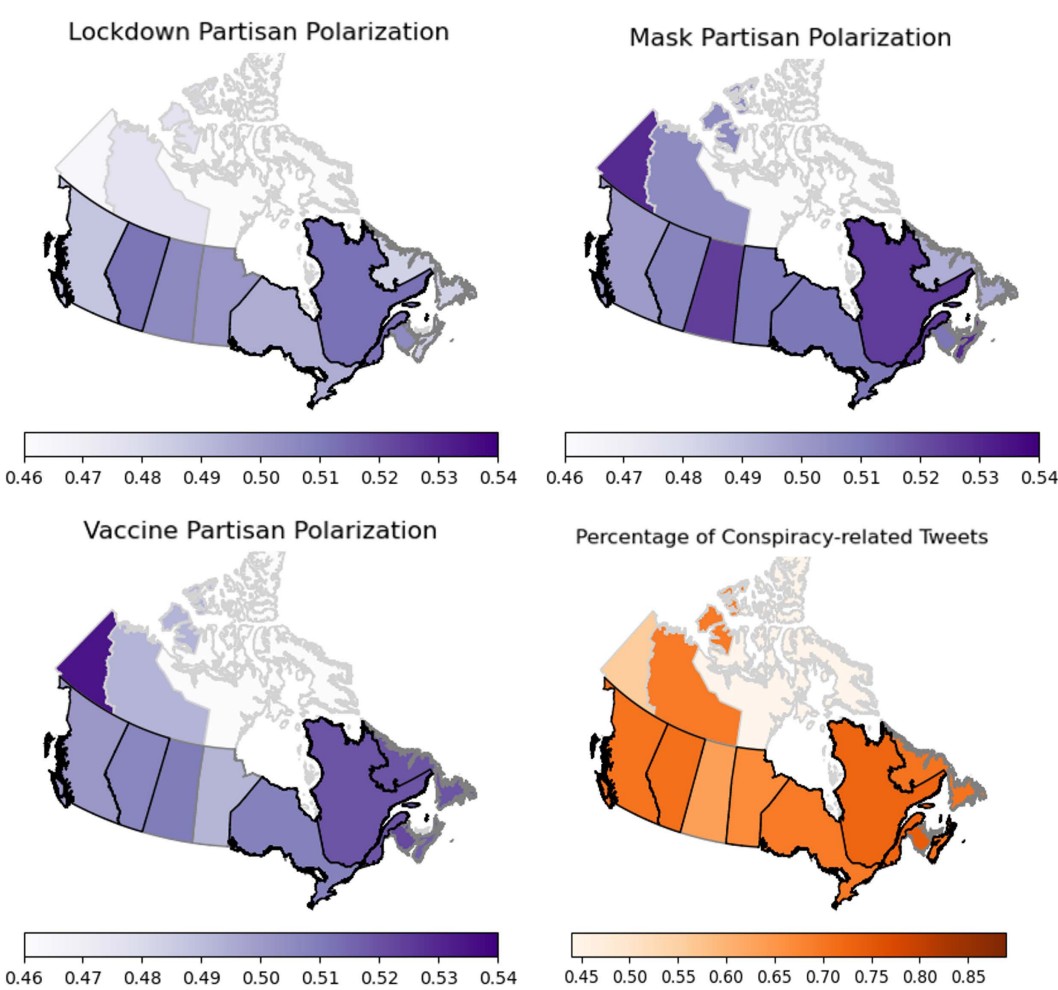

**Fig 8. Regional distribution of partisan polarization in Canada.** A: Lockdown. B: Mask. C: Vaccine. D: % of conspiracy-related tweets. The polarization is measured weekly between October 11, 2020 to January 3, 2021 and the averaged over 12 weeks is used for this plot. We also report the average weekly percentage of conspiracy-related tweets that are posted from users in each region Provinces and territory boundaries are colored based on the number of users we had in our data from those regions, which indicates the support for our measurement: Light-grey for less than 100 users, grey for between 100 and 1,000 users and black for greater than 1,000 users.

the names of predominantly conservative/Republican regions appear in red, predominantly liberal/Democratic regions in blue, and mixed or less definitive regions in purple.

In the United States (Fig 9), conservative states consistently show higher polarization overall and specifically in discussions related to masks and vaccines. The lockdown discourse shows a more mixed pattern, with outliers from both liberal and conservative states; namely Idaho, Alabama, and Arkansas showing the least polarization, and Delaware (ranked 1), Colorado (ranked 17), and New Jersey (ranked 15) showing higher values. However, the general relationship between state partisanship and polarization remains evident, with predominantly liberal states like Vermont (ranked 41) and Massachusetts (ranked 43) displaying less polarization than conservative states like Mississippi (2), North Dakota (3), and Oklahoma (4).

In Canada (Fig 9), Alberta, a conservative province, exhibits higher polarization compared to Ontario and British Columbia (provinces with more social media users in our dataset). Quebec ranks highest overall, though we acknowledge

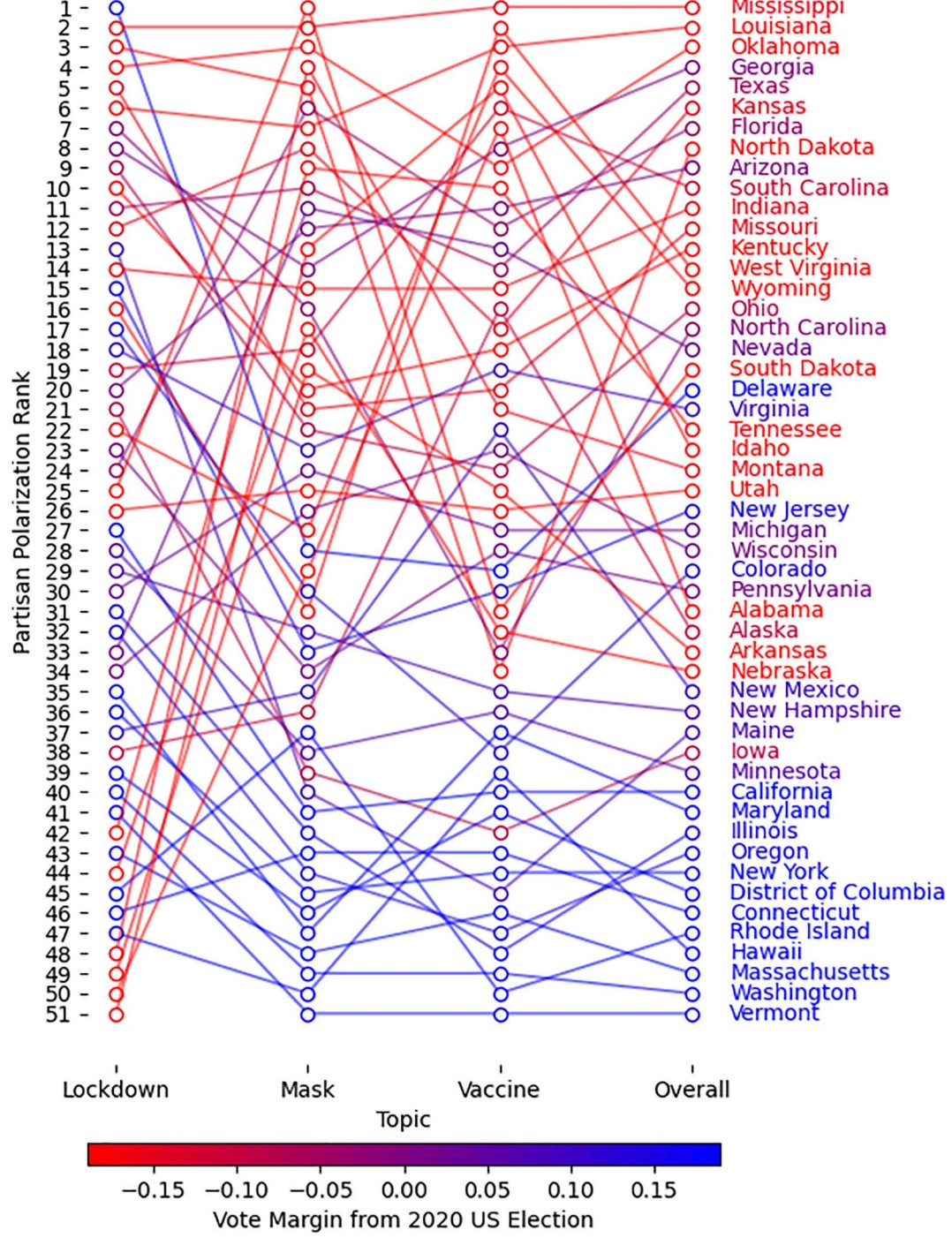

**Fig 9. Ranking of American states partisan polarization per topic and overall.** Ranking of 1 signifies the highest average weekly polarization between October 11, 2020 to January 3, 2021 (12 weeks). State names are colored according to the colorbar (red to blue), based on the vote margin for the conservative party from the 2020 United States Presidential Election (Conservative Party: Republican Party; Liberal Party: Democratic Party).

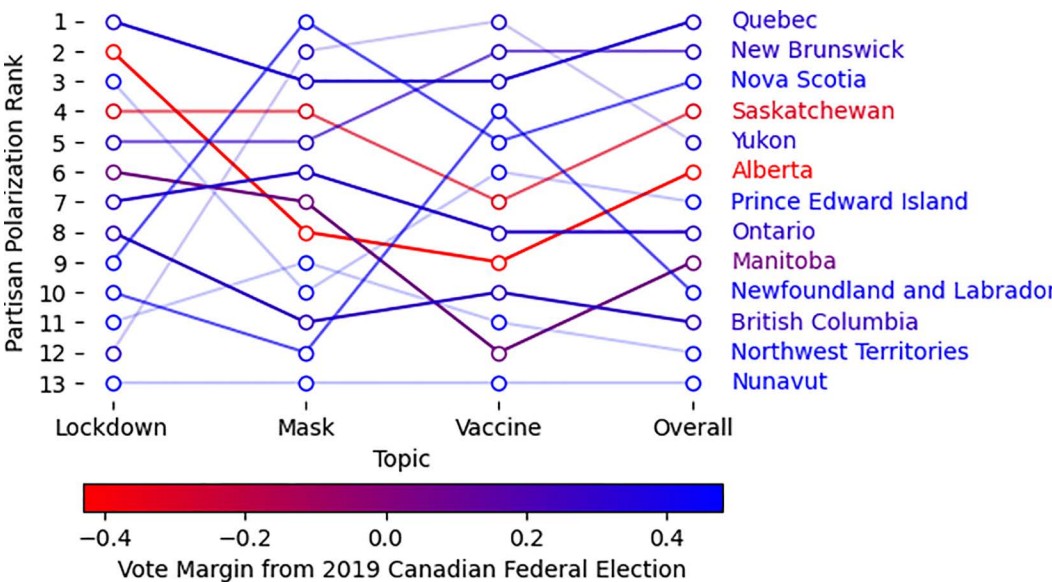

**Fig 10. Partisan polarization ranking of Canadian provinces and territories per topic and overall.** A ranking of 1 signifies the highest average weekly polarization between October 11, 2020 to January 3, 2021 (12 weeks). Province or territory names are colored (red to blue) based on the vote margin for the conservative party family from Canada's 2019 Federal Election (Liberal Party Family: Liberal, New Democratic Party, Green; Conservative Party Family: Conservative, People's Party). Line colors have a transparency to reflect the support for the measurement, based on the number of users in that region.

a limitation of our English-focused analysis in this predominantly French-speaking province, which experienced significant pandemic-related polarization including violent protests [49].

Finally, the correlation between polarization and partisan vote margin is more clearly illustrated in Fig 11A. We find strong, significant correlations in the US between Republican vote share and polarization around masks and vaccines, though not for lockdowns. For Canada, these associations are not statistically significant, partly due to the smaller number of regions analyzed.

Importantly, we observe that **vaccine polarization is strongly negatively correlated with actual vaccination rates across American states** (Fig 12). This aligns with the well-documented negative correlation between Republican vote share and vaccination rates [50]. Using official CDC vaccination data from a comparable period after vaccines became available (October 11, 2021 to January 3, 2022), we confirm a strong negative correlation ($r = -0.77$, $p < 0.001$) between vaccine polarization and vaccination rates. While we cannot establish causality among conservative voting patterns, polarization scores, and vaccination rates, our results suggest polarized discourse played a significant role in shaping the heterogeneous vaccination landscape across the U.S. We did not observe a similar pattern in Canada, due to small sample size and the implementation of vaccine mandates.

## Temporal variation in partisan polarization

Our analysis of daily partisan polarization at the national level (Fig 13) reveals rapid daily fluctuations. This aligns with our hypothesis that language adapts quickly in anticipation of or response to specific events, as compared to timescales in other topic tracking studies, e.g., [51]. In particular, we examined two types of events: 1) pre-selected political and vaccine-related events (Table 8, shown as vertical lines in Fig 13) and 2) highly polarized events detected through analysis of the highest peaks in polarization (Tables 9 and 10, shown as red circles in Fig 13).

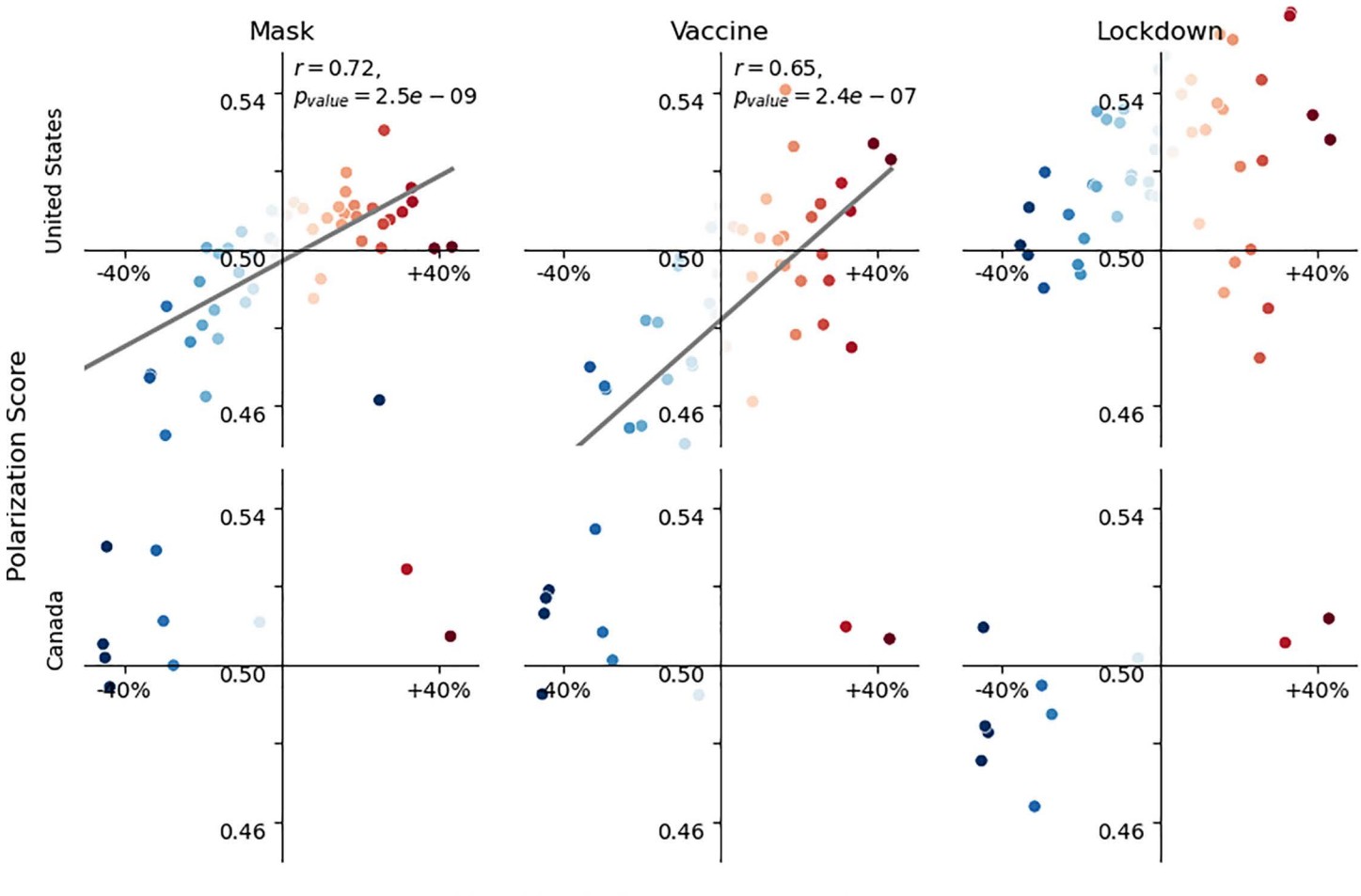

**Fig 11. Correlation between polarization score and vote margin for the conservative party.** Colors (blue to red) are conservative party vote margin (same as Fig 9 and Fig 10). Significant correlation between Polarization Score and Vote Margin is found for the US discourse on masks and on vaccines for which the respective Pearson *r* correlation and *p*-value is shown.

While direct causal evidence is not available, we find that **many of the largest polarization peaks coincide with highly contentious events** specific to each country's context. Below, we analyze these peaks for each country.

**3.2.1. United States polarization timecourse.** The left column of Fig 13 reports the daily polarization measured for the three key topics of lockdowns, masks, and vaccines.

For lockdown-related discourse (Fig 13A), significant polarization peaks occurred around the 2020 Presidential Election. On November 1, 2020 (second-largest peak), President Trump made controversial claims framing the election as a choice between "deadly lockdown measure" supported by Biden or an efficient end to the COVID-19 crisis with a safe vaccine [52]. Trump's campaign consistently characterized lockdowns as tyrannical and economically repressive. Trump also made other similar claims on Twitter (X) during this period, e.g., when he said (sic): "Biden wants to LOCK DOWN our Country, maybe for years. Crazy! There will be NO LOCKDOWNS. The great American Comeback is underway!!!" [53].

Regarding masks (Fig 13C), we observe peaks coinciding with contentious debates. While no external event was identified on October 31, 2020 (highest peak), we found highly polarized social media discourse, with Democrats sharing

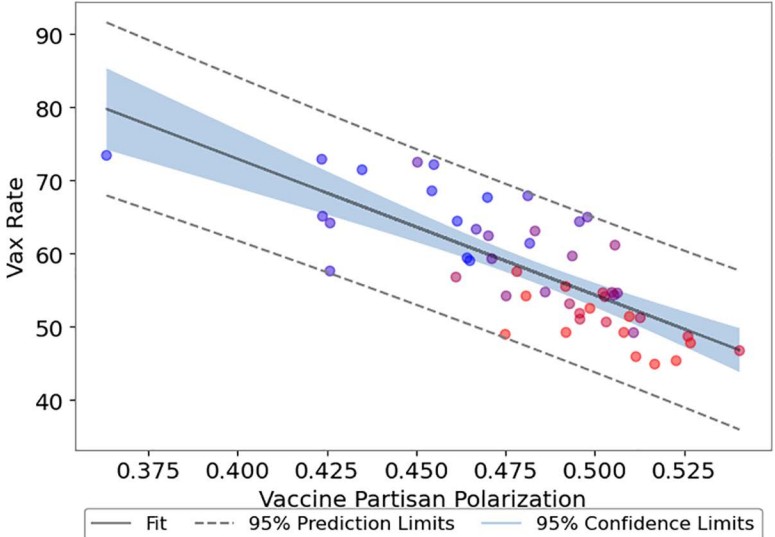

**Fig 12. Relation between vaccines polarization and vaccination rates in the United States.** Color (blue to red) is again the respective conservative party vote margin from the 2020 **U**.S. Presidential Election. The correlation is **−0.77** with CI = [−0.86, −0.62] (n = 51, p = 6.97e-11).

messages like "RT @JoeBiden: Be a patriot. Wear a mask" and Republicans responding with strong opposition "RT @ RealBrysonGray: There's literally nothing patriotic about being so scared of a virus with a 99.9...". This is possibly an example of influencer post-driven, rather than real world event-driven polarization. Another significant peak occurred on November 14, 2020, after Biden proposed mandatory mask mandates and South Dakota Governor Kristi Noem announced her opposition.

Finally, for vaccines (Fig 13E), polarization peaks aligned with key vaccine-related events. The second-largest peak occurred on December 22, 2020, the day after Biden received his first COVID-19 vaccine [54]. While both partisan groups shared the news (most retweeted tweet was "RT @JoeBiden: Today, I received the COVID-19 vaccine. To the scientists and researchers who worked tirelessly to make this possible - than…"), the framing differed significantly — Biden supporters celebrated the moment (e.g., "RT @YAFBiden: And just like that, @JoeBiden has received the COVID-19 vaccine!"), while opponents emphasized Trump's role in Operation Warp Speed (e.g., "RT @ TheLeoTerrell: Finally a @JoeBiden confession. He finally gave credit to @realDonaldTrump and #OperationWarp-Speed. It's about time.")

**3.2.2. Canadian polarization timecourse.** The right column of Fig 13 shows the daily polarization measured for the three key topics of lockdowns, masks, and vaccines. The pre-selected events in the Canadian timeline (Table 8) are marked as vertical lines in the figure.

In Canada's lockdown discourse (Fig 13B), the highest peak occurred on October 17, 2020, coinciding with a large Toronto anti-mask protest against COVID-19 measures. The second-highest peak (November 29, 2020) aligned with a Calgary mask measures protest reported in national news [55].

For masks (Fig 13D), the highest polarization peak appeared on November 14, 2020, concurrent with the US peak, suggesting potential cross-border influence of U.S. partisan discourse on Canadian polarization.

Regarding vaccines (Fig 13F), the two highest peaks occurred on December 20 and 23, 2020, coinciding with the US distribution of the Moderna COVID-19 vaccines and Health Canada's approval of the Moderna vaccine [56], respectively. The U.S. event sparked discussions in Canada about vaccine prioritization and availability [57], with top retweets by liberal party family users focused on news of the Republican politicians being first in line for the vaccine, while

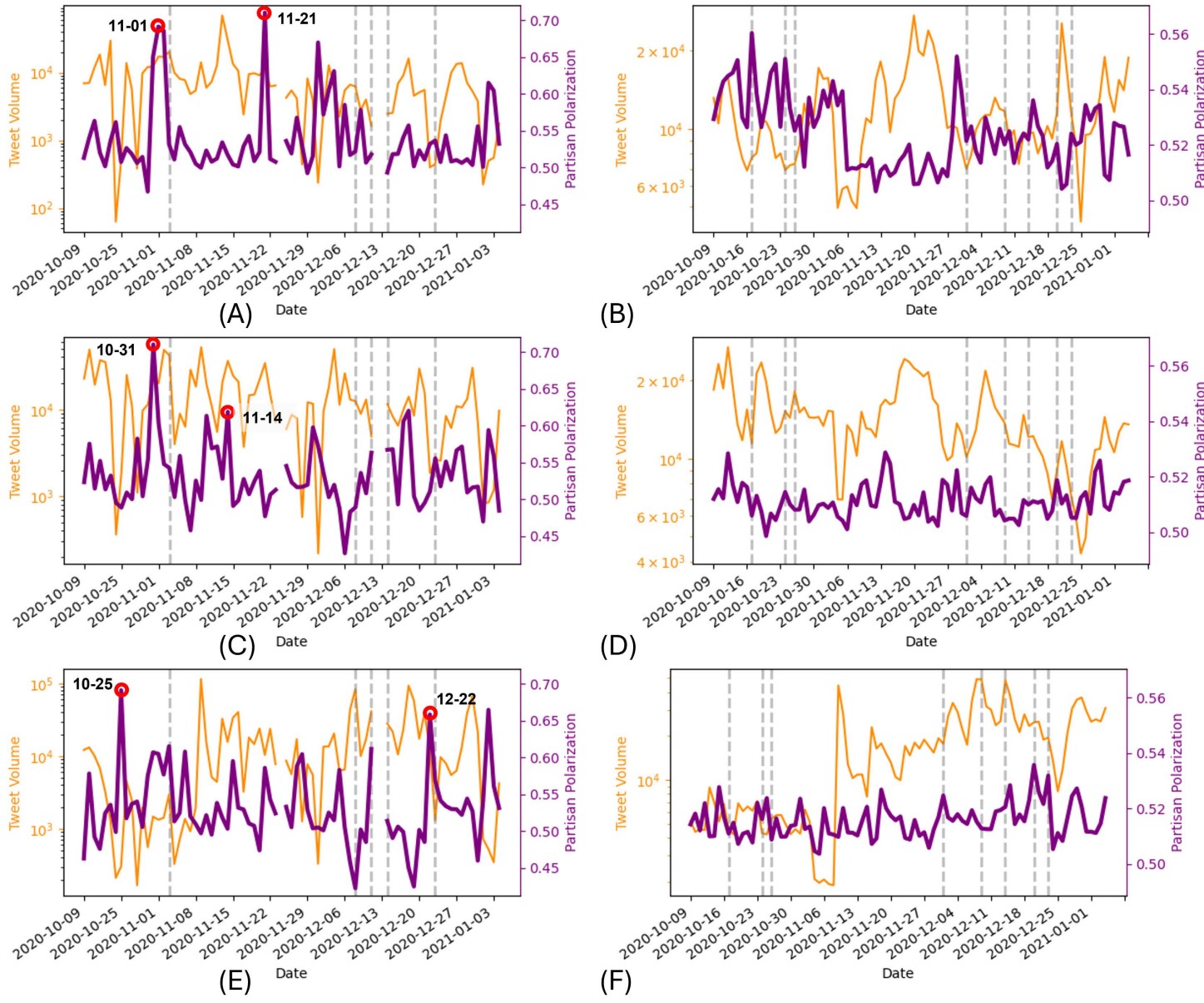

**Fig 13. Daily trends of partisan polarization in the United States and Canada from October 9, 2020 to January 3, 2021.** A: US Daily Lockdown Polarization. B: Canadian Daily Lockdown Polarization. C: US Daily Mask Polarization. D: Canadian Daily Lockdown Polarization. E: US Daily Vaccine Polarization. F: Canadian Daily Vaccine Polarization. The vertical dashed lines denote pre-selected political and vaccine-related events as explained in the text. In addition to the polarization measure (purple line), we also report the tweet volume, in log-scale, on the corresponding topic (yellow line) per day which denotes the size of support for our measurement.

conservative party family users retweeted more diverse anti-vaccine sentiment. The Canadian approval few days later generated strong polarized sentiments supporting and opposing the decision as shown with the top retweets.

**3.2.3. Aggregate polarization.** To evaluate the overall responsiveness of our measure to polarizing events, we computed average polarization scores across topics (Fig 14A,B) and performed event-triggered averaging. This aggregate analysis (Fig 14c) confirms a rapid, symmetric rise-and-decay profile around polarization peaks, occurring on timescales of days.

**Table 8. Major political and pandemic-related events in each country.**

| Date | Event | Country |
|------|-------|---------|
| Nov. 3 | US National Election | US |
| Oct. 24 | BC General Election | Canada |
| Oct. 26 | Saskatchewan General Election | Canada |
| Dec. 8 | States resolve controversies | US |
| Dec. 9 | PHAC approves Pfizer vaccine | Canada |
| Dec. 11 | FDA approves vaccines | US |
| | Electoral votes submitted | US |
| Dec. 14 | Vaccination begins | Canada |
| Dec. 20 | Moderna vaccine distributed | US |
| | Election votes arrive | US |
| Dec. 23 | PHAC approves Moderna Vaccine | Canada |

Table contains dates in the early COVID-19 pandemic (2020) such as when the FDA (U.S. Food & Drug Administration), and PHAC (Public Health Agency of Canada) approved vaccines for both the United States and Canada.

**Table 9. US polarization peaks and their corresponding events.**

| Topic | Date | United States Polarizing Event |
|-------|------|-------------------------------|
| Lockdown | Nov. 1 | Viral tweet by Trump |
| | Nov. 21 | Unidentified topic |
| Masks | Oct. 21 | Viral tweet by Trump |
| | Nov. 14 | Biden proposes mandates |
| Vaccines | Dec. 20 | Moderna vaccine distributed |
| | Dec 22 | Biden gets vaccinated |

For each topic, we analyzed the two highest peaks, and inferred the content discussed on those peaks.

**Table 10. Canadian polarization peaks and their corresponding events.**

| Topic | Date | Canada Polarizing Event |
|-------|------|------------------------|
| Lockdown | Oct. 17 | Toronto Mask Measures Protest |
| | Oct. 29 | Calgary Mask Measures Protest |
| Masks | Oct. 12 | Unidentifed topic |
| | Nov. 14 | PHAC recommends masks |
| Vaccines | Dec. 20 | Moderna distributed in U.S. |
| | Dec. 23 | PHAC approves Moderna |

For each topic, we analyzed the two highest peaks, and inferred the content discussed on those peaks.

## Relationship between conspiracy and polarization

Finally, we explored the relationship between conspiracy discourse and polarization using time-resolved measurements of conspiracy-related tweets compared with aggregate daily polarization. The profiles by partisan affiliation (Fig 15A,C)

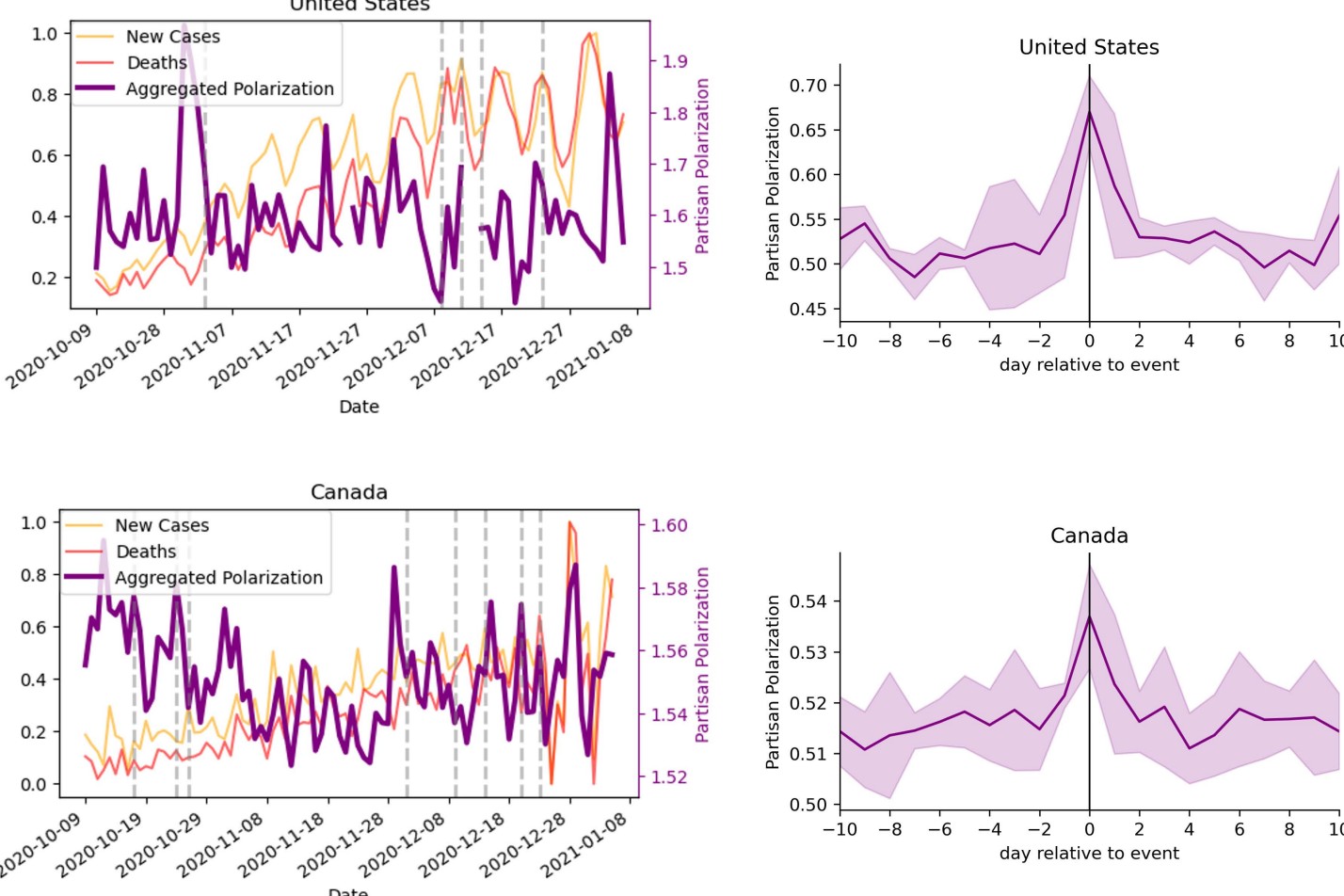

**Fig 14. Daily aggregated partisan polarization.** For the **U.**S. (A, B) and Canada **(C, D)**, polarization is aggregated over topic by averaging over the values. We show pandemic-related new cases and deaths in background for reference. Event-triggered average polarization for identified events are listed in Table 9 and Table 10 respectively. Shaded region denotes standard deviation over the 5 events for each country.

show that conservative partisans in both countries tweet conspiracy-related content at higher rates than progressive partisans.

When correlated with aggregate daily polarization (Fig 15 B and D, we find a small but significant negative correlation for the United States, while no significant correlation was observed for Canada.

## Discussion

We investigated regional and event-triggered variation in partisan division within social media debates surrounding COVID-19 public health measures across American states and Canadian provinces. Our computational analysis quantified partisan polarization by analyzing the language used in millions of tweets from users affiliated with different political parties. We focused specifically on Twitter (X) discussions related to three key public health interventions during the early phases of the COVID-19 pandemic: lockdowns, masks, and vaccines, while also tracking the volume of conspiracy-related tweets.

Our analysis explored both the geographic heterogeneity of polarization and identified political events that likely influenced public opinion over time. We found distinctive patterns between the United States and Canada, highlighting important cross-national differences in how partisan polarization manifested online during the pandemic.

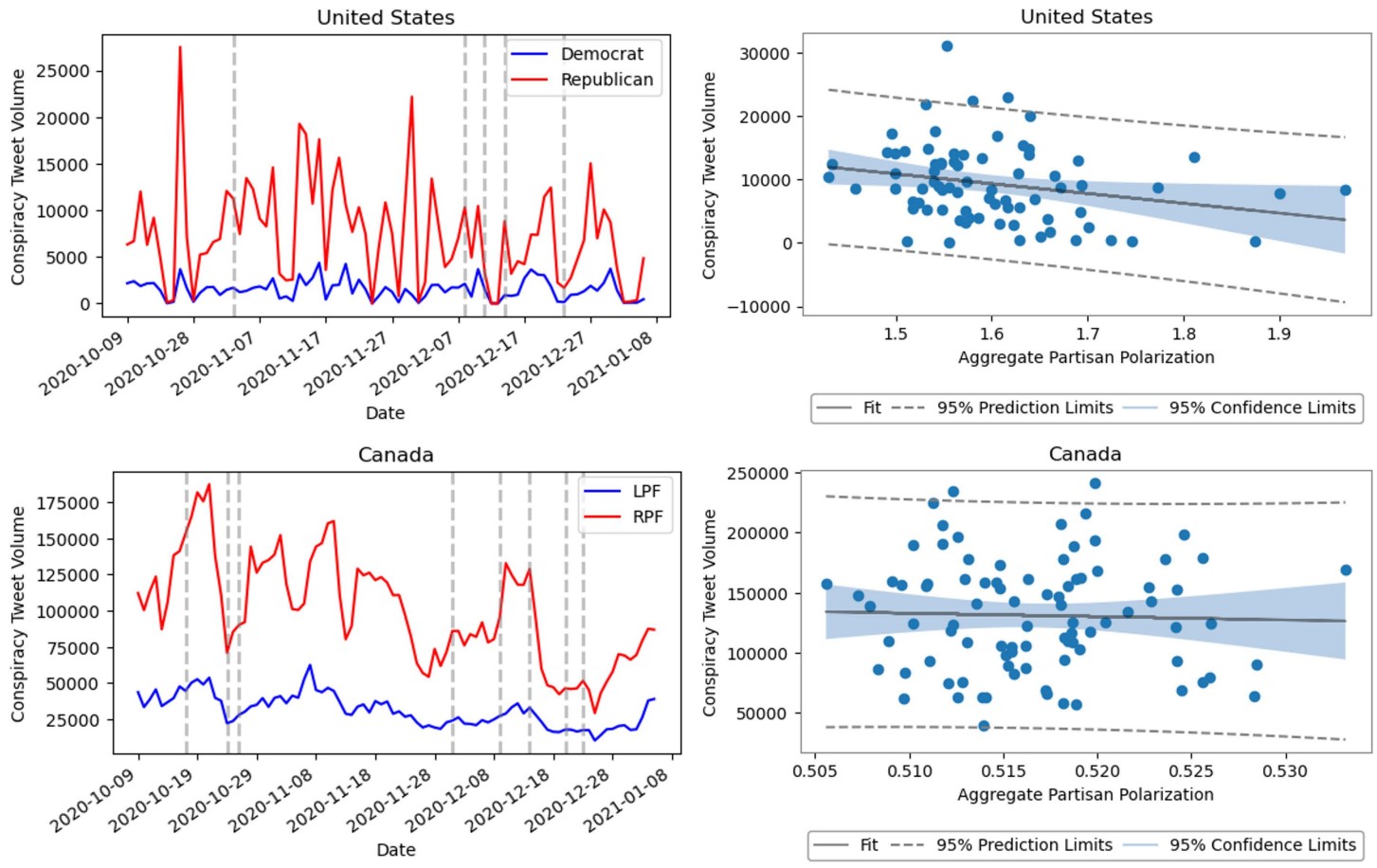

**Fig 15. Relation between the volume of conspiracy related content and the observed partisan polarization in the United States and Canada.** In the left column, we report the volume of conspiracy related tweets posted by users for the United States (A: affiliated with the Democrat and Republican party) and Canada (C: affiliated with the liberal (left) Party Family (LPF)—Liberal, New Democratic Party, Green, and the conservative (right) Party Family (RPF)—Conservative, People's Party for Canada). On the right, we show the relation between daily partisan polarization summed over the different topics and the overall volume of conspiracy tweets. In the United States **(B)**, we find there is a statistically significant correlation **−0.247** with CI=[−0.448,-0.023] (n = 88, p = 0.031) between these measures. In Canada **(D)**, we find that there is no statistically significant correlation of **0.023** with CI=[−0.187,0.231] (n = 88, p = 0.831) between these measures.

## United States

In line with previous research ((*e.g.*, [17,58–60])), we found that more right leaning states and provinces exhibited greater partisan divisions around COVID-19 on Twitter (X), particularly concerning mask mandates and vaccine distribution. However, we extended beyond these studies to characterize the geographic heterogeneity and temporal evolution of polarization, connecting features in our polarization metric to real-world events. Our analysis revealed a strong negative correlation between partisan polarization and future vaccination rates, and a moderate negative correlation between the temporal profiles of conspiracy-related tweets and aggregate polarization in the U.S.—patterns not observed in Canada.

The early phases of the COVID-19 pandemic in the United States prompted online discussions that reflected strong regional variation in partisan support [25,61–64]. Rather than finding uniformly low state-specific polarization with internally homogeneous semantics within conservative and liberal states, our analysis confirmed significantly higher

polarization in conservative states. Republican vote margins had a significant positive effect on polarization in discussions concerning masks and vaccines, even after controlling for other factors.

This main finding is consistent with previous studies suggesting conservative states exhibited higher polarization (Fig 11) in response to public health interventions compared to liberal counterparts [10,62]. However, this pattern was not universal. Delaware, a liberal state, exhibited unexpectedly high polarization, likely due to strict public health measures implemented by the governor in response to rapidly increasing cases during the first pandemic wave [65–67]. Conversely, several conservative states like Arkansas, Alabama, and Idaho showed low polarization levels, possibly due to more unified opposition to restrictive COVID-19 measures like mask mandates [68].

While our analysis did not identify a single explanation for the relationship between conservative politics and higher polarization, state-specific trajectories provide insights. Mississippi, North Dakota, and Oklahoma experienced political decisions—such as mask mandates—early in the pandemic that led to public resistance despite rising COVID-19 cases [61,69,70]. In Mississippi, the governor's decision to lift the statewide mask mandate in late September 2020 appears to have contributed to heightened polarization [68,69]. Similar patterns emerged in North and South Dakota and Oklahoma, where initial hesitancy to enforce mask mandates apparently led to increased partisan divisions [ 10,68,71,72,73–75].

One plausible explanation for our main finding is that as pandemic severity increased, it strained the initially more uniform opposition to restrictive health measures in conservative states, leading to higher polarization levels [68,76,77]. This represents a distinct source of polarization compared to states with more equal distributions of partisans across competitive districts.

Our approach also revealed how polarization varied temporally across the three topics—lockdowns, masks, and vaccines. All three exhibited similar baseline polarization levels during our study period (between the second and third pandemic waves), with large, sharp positive deviations. Lockdowns showed fewer deviations, with low correlation to Republican party vote share, suggesting they did not serve as a meaningful indicator of partisan opposition. Masks and vaccines, however, displayed numerous sharp polarization peaks that we could often associate with specific real-world events.

For example, South Dakota initially experienced very high mask-related polarization coinciding with medical authorities promoting mask-wearing despite Governor Kristi Noem's opposition [68,78]. Her public opposition to Biden's mask mandate suggestions created a notable polarization peak. Similarly, North Dakota displayed early increased polarization regarding conspiracy theories [61], potentially exacerbated by a posthumous electoral win of a Republican candidate who died from COVID-19 [79–81].

The strong negative correlation between vaccine polarization and vaccination rates in the U.S. demonstrates that states with higher vaccination rates were less polarized around this issue [82]. While the exact causal mechanism remains unclear, factors such as education and political ideology, which also have strong geographic dependencies, likely played significant roles [ 60]. Higher education levels are generally associated with greater vaccine acceptance and trust in vaccine safety [83], while Democrats typically showed higher trust in COVID-19 vaccines and were early adopters compared to Republicans [84].

Overall, the high polarization levels observed in the United States relative to Canada indicate a more divided society. Our study provides new insights by demonstrating that partisan divisions correlate with regional heterogeneity in social media discourse, particularly during salient political events concerning health measures. This discourse reflects the pandemic timeline evolution—initially centered on stay-at-home orders, then mask mandates, and later transitioning to vaccines as they became available.

### Canada

Our Canadian results suggest that partisan divisions also influenced public responses to COVID-19 measures [85], though to a lesser degree than in the U.S. We found a similar but much weaker association between polarization and conservative political leaning, with conservative provinces like Alberta and Saskatchewan experiencing higher polarization

during stricter lockdown measures than more liberal counterparts such as British Columbia and Ontario [49,86]. Polarization levels varied across smaller and medium-sized provinces as well, measuring relatively high for New Brunswick and low for Nunavut, where COVID-19 infection rates remained relatively low throughout the pandemic (the sole COVID-free jurisdiction in North America until November 2020) [87,88].

Quebec presents a particularly interesting case in our analysis. Our results confirmed that Quebec had the highest level of partisan division over vaccines, alongside the highest reported COVID-19 incidence in Canada during the first and second pandemic waves [89]. Quebec's unique, more restrictive approach to pandemic management is reflected in our results. After temporarily relaxing several restrictions in summer 2020, Quebec again became the pandemic epicenter that fall [90], leading to reinstatement of strict control measures and bans on public demonstrations following significant anti-government protests [91,92]. These events coincided with increased online conflicts, promoted by Canadian far-right populist rhetoric and conspiracy theories on Twitter (X) [93].

Importantly, most COVID-related conversations in Canada were heavily influenced by U.S. discussions, with Canadians retweeting American vaccine-related content eight times more frequently than Canadian content during our study period [94,95]. Like in the U.S., vaccine hesitancy was linked to political affiliation in Canada, with Conservative Party supporters more likely to refuse vaccination [96].

As with the U.S., the Canadian polarization time course exhibits spikes around key events such as protests against lockdown measures, mask mandates, and vaccine roll-outs. However, unlike the U.S., we did not observe a negative correlation between polarization and conspiracy-related tweet volume. This contrasts with [94], who found reduced negative sentiment in Canadian vaccine-related tweets between January and December 2020. The relationship between polarization and sentiment is complex, with long-term trends likely driven by processes beyond mere discussion volume around conspiracy theories [97].

The relatively lower influence of polarization on vaccine attitudes in Canada may be attributed to the country's more widespread vaccine mandates [98,99]. This, combined with higher levels of trust in politicians [100] and social capital [101,102], may have contributed to broader acceptance of COVID-19 health interventions [95,96]. Indeed, there was a rare "cross-partisan consensus" among Canadians regarding emergency measures in the early stages of the pandemic [103]. This consensus, however, was not mirrored on social media, where conspiracy theories widely circulated [24,95]. Overall, our results indicate that online discussions surrounding lockdowns, masks, and vaccines did mirror polarization patterns shaped by regional reactions to events and circumstances specific to Canadian provinces, albeit to a lesser degree than in the U.S.

## Limitations

While our method offers valuable insights, several limitations must be acknowledged. First, we measured partisan polarization only through the proxy of semantic similarity, which may in certain cases obscure signals not captured by semantic embedding representations. Second, in the Canadian context specifically, we categorized users into liberal (left) and conservative (right) party families. During manual annotation of Twitter (X) profiles, we encountered few users who identified as Bloc Québécois supporters and excluded them from analysis. Additionally, our classification into partisan groups relies on self-reported information, which may not be entirely accurate.

Third, our analysis is based on Twitter (X) data, which may not fully capture the views and sentiments of the broader American and Canadian public. Twitter users are more politically engaged and often more partisan than the general population, which means our results reflect dynamics within a vocal subset of online political discourse. Fourth, we restricted our analysis to English-language tweets, which in Canada means we primarily captured perspectives of either anglophones or bilingual francophones — a potential source of bias. For example, the high polarization observed in Quebec regarding COVID-19 measures may be influenced by this language limitation.

Fifth, geographic attribution of users depends on self-reported location fields processed through tools such as Open Street Map. While this is a reasonable approach, it is subject to inaccuracies if users provide false, incomplete, or

humorous locations. Furthermore, the use of VPNs or fake accounts (including bots) may introduce additional error. Automated accounts can contribute disproportionately to partisan discourse by generating large volumes of ideologically slanted content or amplifying existing narratives, potentially inflating the apparent semantic distance between political groups.

Sixth, there are differences in keyword strategies used between the U.S. and Canadian datasets. The U.S. keyword list was dominated by election-related terms, then filtered for COVID-19 terms. By contrast, the Canadian keyword list incorporated province names and explicit COVID-related identifiers, which likely offered stronger pandemic relevance. This asymmetry may have contributed to cross-national differences in patterns of observed polarization, and this should be considered when interpreting the comparative results.

Seventh, there are temporal mismatches between our Twitter dataset (collected in late 2020) and regional vaccination rates, which occurred primarily in 2021. As a result, correlations between polarization in vaccine-related discussions and later vaccination uptake should be understood as suggestive rather than causative. They reflect potential predictive relationships between early partisan discourse and subsequent health behaviors, but these interpretations should remain cautious.

Finally, while several of our analyses rely on correlations, these results do not imply causation. The relationship between polarization and public health measures is complex and multidimensional, influenced by numerous social, political, and cultural factors that extend beyond the scope of our computational analysis.

## Conclusion

Our computational approach to measuring partisan polarization has revealed important insights into how political divisions manifested during the COVID-19 pandemic across the United States and Canada. By analyzing millions of tweets and tracking semantic differences between partisan groups across different regions and topics, we have demonstrated that polarization is not uniform but varies significantly by geography, political leaning, and public health topic, with substantial differences between the two countries.

In the United States, we found compelling evidence that conservative states exhibited higher polarization around mask mandates and vaccines, with partisan division strongly negatively correlated with vaccination rates. The temporal patterns in our polarization metric successfully captured real-world political events, providing a powerful lens through which to understand how public discourse evolves in response to political messaging and policy decisions. The strong relationship between Republican vote share and polarization on masks and vaccines—but not lockdowns—suggests that partisan identity became increasingly attached to specific public health measures as the pandemic progressed.

Canada, while showing similar patterns, demonstrated significantly lower overall polarization levels. The weaker association between conservative politics and polarization in Canada highlights how different political environments and institutional structures can moderate partisan divisions even during global crises. The relative consensus at federal and provincial levels regarding the effectiveness of public health measures appears to have created a more unified public discourse, even as some regions like Quebec experienced higher levels of division.

The cross-national comparison offers particularly valuable insights: while both countries experienced social media polarization, the U.S. showed stronger correlations between polarization, conspiracy theories, and public health outcomes such as vaccination rates. This suggests that the impact of online polarization on public health behaviors may be mediated by broader political and institutional factors. The Canadian case demonstrates that even with exposure to the same social media platforms and significant content sharing across borders (with Canadians retweeting American content eight times more than Canadian content), the local political environment remains crucial in determining how online polarization translates to public opinion and behavior.

These findings have important implications for both researchers and policymakers. For researchers, our semantic similarity approach to measuring polarization offers a robust methodology that can be applied to other contexts and issues.

For policymakers and public health officials, our work underscores the importance of understanding regional and partisan differences when designing communication strategies during health emergencies. The ability to identify particularly polarizing topics and track how polarization evolves in response to specific events could help authorities craft more effective messaging that resonates across partisan divides.

Future research should explore the causal mechanisms behind the correlations we identified, particularly the relationship between online discourse and offline behaviors such as vaccine uptake. Additionally, the role of cross-border information flows in shaping polarization deserves further investigation, especially as online communities increasingly transcend national boundaries while still being influenced by local political contexts. As our analysis demonstrates, social networks clearly contributed to the diffusion of partisan opinions during the COVID-19 pandemic, but quantifying their precise impact on polarization and subsequent public health outcomes remains an important challenge for future work.

## Supporting information

**S1 Fig. Ranking of American states by partisan polarization per topic.** Ranking of 1 signifies the highest average weekly polarization between October 11, 2020 to January 3, 2021 (12 weeks). State names are colored based on the party ratio from the 2020 United States Presidential Election, where more blue means more users voted for the Democratic Party and more red means more users voted the Republican Party. We can see that red states are mostly ranked higher than blue states.
(PNG)

**S2 Fig. Ranking of Canadian provinces and territories by partisan polarization per topic.** A ranking of 1 signifies the highest average weekly polarization between October 11, 2020 to January 3, 2021 (12 weeks). Province or territory names are colored based on the party ratio from Canada's 2021 Federal Election, where more blue means more users from the liberal (left) party family (Liberal, New Democratic Party, Green), and more red means more users from the conservative (right) party family (Conservative, People's Party).
(PNG)

**S3 Fig. Relation between vaccine polarization and vaccination rates in Canada.** We remove Nunavut as an outlier because of its very small population. The correlation is **0.74** with CI = [0.31, 0.92] (n = 12, p = 0.004). The Vaccine Partisan Polarization for each province or territory is computed weekly and averaged over 12 weeks from October 11, 2020 to January 3, 2021. Official vaccination rates for different regions are obtained from Statistic Canada. The Vaccination Rate is also averaged weekly for the similar period of time a year into future to be after the vaccines were rolled out, i.e., October 11, 2021 to January 3, 2022. Color for the scatter plots is determined by the respective party ratio from the 2021 Canadian federal election. While we get strong positive correlation with vaccination rate, it is over a relatively low number of points. In Canada, vaccines were mandated, requiring vaccine passports to be served in public areas. We assume that vaccine partisan polarization increases, as people are not happy with being forced to be vaccinated, but most of the population still are vaccinated. However, with the few points, we do not have a definite conclusion for this result.
(PNG)

**S4 Fig. Weekly trends of partisan polarization in the United States.** We present them for A: Highest ranked states overall (Mississippi). B: Lowest ranked state (Vermont). C: Highest ranked liberal state (Delaware). D: Lowest ranked conservative state (Iowa). We report the average death rate (red dotted line) per week and report the correlation between the topic-specific correlation with the death rate in the brackets in the legend.
(PNG)

**S5 Fig. Weekly trends of partisan polarization in Canada for the top 4 largest provinces.** A: Alberta. B: British Columbia. C: Ontario. D: Quebec. We report the average death rate (red dotted line) per week and report the correlation between the topic-specific correlation with the death rate in the brackets in the legend.
(PNG)

**S6 Fig. Daily aggregate polarization v.s. COVID-19 new cases and deaths for the United States and Canada.** Here, we investigate the aggregated polarization over time for each country and how it relates to the reported number of New Cases and Deaths for COVID-19. To compute the daily aggregate polarization measure, we employ a weighted sum of each topic's polarization, considering the percentage of each topic's tweets within that day's volume of COVID-19-related tweets. The correlation coefficient are **−0.196** for the United States with CI = [−0.403,0.031] (n=88, p=0.090) and **−0.044** for Canada with CI = [0.251,0.167] (n=88, p=0.681).We observe that *polarization is not correlated with the severity of the pandemic, in both the United States and Canada.*
(PNG)

**S1 Table. Correlation matrix between topic polarization and external data in the United States.** Bolded means $p < 0.001$. Italicized means $p < 0.01$. Underline means $p < 0.05$. Background color of green or red signifies the positive or negative correlation for significant p-values only.
(PDF)

**S2 Table. Correlation matrix between topic polarization and external data in Canada.** Bolded means $p < 0.001$. Italicized means $p < 0.01$. Underline means $p < 0.05$. Background color of green or red signifies the positive or negative correlation for significant p-values only.
(PDF)

## Author contributions

**Conceptualization:** Jean-Francois Godbout, Reihaneh Rabbany.

**Data curation:** Zachary Yang, Anne Imouza, Cecile Amadoro, Gabrielle Desrosiers-Brisebois, Sacha Levy.

**Formal analysis:** Zachary Yang, Anne Imouza, Maximilian Puelma Touzel.

**Funding acquisition:** Jean-Francois Godbout, Reihaneh Rabbany.

**Investigation:** Zachary Yang, Anne Imouza, Maximilian Puelma Touzel, Cecile Amadoro, Gabrielle Desrosiers-Brisebois, Kellin Pelrine.

**Methodology:** Zachary Yang, Kellin Pelrine.

**Resources:** Sacha Levy.

**Software:** Zachary Yang.

**Supervision:** Jean-Francois Godbout, Reihaneh Rabbany.

**Validation:** Zachary Yang, Anne Imouza, Maximilian Puelma Touzel, Gabrielle Desrosiers-Brisebois, Jean-Francois Godbout.

**Visualization:** Zachary Yang.

**Writing – original draft:** Zachary Yang, Anne Imouza, Cecile Amadoro, Gabrielle Desrosiers-Brisebois.

**Writing – review & editing:** Anne Imouza, Maximilian Puelma Touzel, Cecile Amadoro, Gabrielle Desrosiers-Brisebois, Kellin Pelrine, Jean-Francois Godbout, Reihaneh Rabbany.

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
