## [Decision Letter · Decision Letter 0]

15 Aug 2025

We look forward to receiving your revised manuscript.

Kind regards,

Biljana Gjoneska

Academic Editor

PLOS ONE

Journal Requirements:

2. In your Methods section, please include additional information about your dataset and ensure that you have included a statement specifying whether the collection and analysis method complied with the terms and conditions for the source of the data.

4. We note that Figure 2, Figure 3, Figure 7 and Figure 8 in your submission contain map images which may be copyrighted. All PLOS content is published under the Creative Commons Attribution License (CC BY 4.0), which means that the manuscript, images, and Supporting Information files will be freely available online, and any third party is permitted to access, download, copy, distribute, and use these materials in any way, even commercially, with proper attribution. For these reasons, we cannot publish previously copyrighted maps or satellite images created using proprietary data, such as Google software (Google Maps, Street View, and Earth). For more information, see our copyright guidelines: http://journals.plos.org/plosone/s/licenses-and-copyright.

1. You may seek permission from the original copyright holder of Figure 2, Figure 3, Figure 7 and Figure 8 to publish the content specifically under the CC BY 4.0 license.

In the figure caption of the copyrighted figure, please include the following text: “Reprinted from [ref] under a CC BY license, with permission from [name of publisher], original copyright [original copyright year].

5. For studies involving third-party data, we encourage authors to share any data specific to their analyses that they can legally distribute. PLOS recognizes, however, that authors may be using third-party data they do not have the rights to share. When third-party data cannot be publicly shared, authors must provide all information necessary for interested researchers to apply to gain access to the data. (https://journals.plos.org/plosone/s/data-availability#loc-acceptable-data-access-restrictions)

Additional Editor Comments:

Esteemed Authors,

Thank you for your patience with the prolonged review process. Your understanding for our voluntary services (aimed entirely for the advancement of the academic community and the science) is much appreciated indeed.

I have finally received the reports from two reviewers with differing areas of research interest (Computer Science and Psychology) but complementary and converging assessments. I am pleased to report that both reviewers find some merit in your work titled "Regional and temporal patterns of partisan polarization during the COVID-19 pandemic in the United States and Canada" (ID: PONE-D-25-11475), highlighting that the study is "relevant, interesting and insightful" (Rev 1), as well as "timely and methodologically innovative" (Rev 2). However, both reviewers also identify some major shortcomings, and are unified in their observation that the study is pending major revision before even being considered for proceeding further. The manuscript requires rework in all major areas as follows:

- The introductory part requires better positioning of your study within the existing scholarly literature on the topic, especially with respect to existing LLMs exploring similar subjects (as pointed out by Rev 1), and with respect to the political and temporal context (as pointed by Rev 2).

- The methodological part should be strengthened by including rationale for the method of choice (text similarity) and comparisons with other network-based approaches (as pointed out by Rev 1), as well as clarification (definitions, formal criteria for evaluation) of some of the investigated variables like conspiracy theories and sentiment analysis (as pointed out by Rev 2).

- The results require major rework of nearly all figures that are of unacceptable low quality. Take for example the opening illustration (Fig.1) which offers conceptual design of the study as a way to orient the reader, but it is so overloaded and busy, that the effect is completely the opposite (so it better be excluded). With respect to the rest of the figures, please take care to address all issues raised by Rev 1 as regards this.

- The limitations should include acknowledgments (justifications) of the temporal differences in the period of data collection vs period of vaccination, as well as note of the observations made by Rev 2.

Overall, this is rather ambitious undertake (examining phenomena across large data and populations, embedded in complex political and cultural contexts), that might easily collapse under "its own weight" of going in many different directions but (possibly) leading nowhere outside the expected conclusions (as regards the polarization across party lines that is more pronounced in US than Canada, and more pronounced for RW than LW partisans). This is a tangible risk if the work is not clearly directed, carefully guided and very focused, and written in a corresponding manner. With respect to the language, Rev 2 objects the use of the frequent evaluative language, which should be replaced with informative, concrete and language that is supported by facts.

The reviewers have also identified some more specific issues and carefully listed them in their reports. Please make every effort to address each issue with sufficient quality and in sufficient manner. Also, please note that the revised manuscript might still be considered not finalized or even acceptable for publication, as some additional issues might arise during the next round of revisions by the same or a new set of reviewers (depending on their availability).

With this I conclude my message and send best regards for productive revision of your manuscript.

Sincerely,

Biljana Gjoneska

Reviewer's Responses to Questions

**Comments to the Author**

1. Is the manuscript technically sound, and do the data support the conclusions?

Reviewer #1: Yes

Reviewer #2: Partly

2. Has the statistical analysis been performed appropriately and rigorously?

Reviewer #1: Yes

Reviewer #2: I Don't Know

3. Have the authors made all data underlying the findings in their manuscript fully available?

Reviewer #1: No

Reviewer #2: Yes

4. Is the manuscript presented in an intelligible fashion and written in standard English?

Reviewer #1: Yes

Reviewer #2: Yes

Reviewer #1: This manuscript presents a computational study of online partisan polarization during the COVID-19 pandemic in the U.S. and Canada. Analyzing over 50 million tweets, the authors propose using language model embeddings to measure polarization across regions and time. The findings show stronger polarization in conservative areas—especially in the U.S. The study is relevant, interesting, and insightful. However, the paper could have been organized better, positioned more precisely in the literature, written more clearly, and the figures should have been better for an appropriate evaluation.

My two major concerns are:

- As the authors have noted, the approach relies on text similarity rather than explicit stance or relational structure. Embedding-based measures can miss key contextual cues such as sarcasm, negation, or indirect disagreement. On the other hand, there are other approaches based on investigating the network of user interactions, which may be more insightful as it can consider the actual flow of information. The authors should at least provide a comparison of the benefits and drawbacks of their approach compared to network-based approaches. One study that explores polarization surrounding vaccination discussions on Twitter is given in Cinelli et al., 2021.

> Cinelli, M., De Francisci Morales, G., Galeazzi, A., Quattrociocchi, W., & Starnini, M. (2021). The echo chamber effect on social media. Proceedings of the national academy of sciences, 118(9), e2023301118.

- The authors have presented their approach as novel, but having in mind that language models have been around for a long time now, I think that there have already been other similar applications, such as the one from He et al., 2021, so I think the authors should better position their work in the literature.

> He, Z., Mokhberian, N., Câmara, A., Abeliuk, A., & Lerman, K. (2021). Detecting polarized topics using partisanship-aware contextualized topic embeddings. arXiv preprint arXiv:2104.07814.

My other remarks are:

- Line 108: As I understand, these data-filtering keywords are different from the topic-specific keywords discussed later, so the authors should consider renaming them.

- Line 120: The authors should comment how do they believe the time difference between vaccination rates and Twitter data collection affect the results.

- Lines 149-150: It is unclear whether RoBERTa was fine-tuned only on manually validated tweets or all hashtag-annotated ones.

- Lines 151-152: It is unclear what is evaluated, manual annotations or classification of many tweets based on RoBERTa.

- In Table 1, what exactly are the five runs? Are there five different repetitions of 200 pairs of tweets?

- Line 159: What do the authors mean by using pre-trained NER? NER will need to be combined with other geo-location services, such as those mentioned. Otherwise, I believe the services use an in-house NER system.

Figure 3: Subfigure captions are missing. They are probably the same as in the previous Figure, but they should also be mentioned here.

- Figures: The Figures are of unacceptably low quality, so they must be improved. Many of the axes are not annotated. Figures 4, 5, and 6 have the same name, and it would be clearer for the reader if they had unique names. What is CV in Figure 4?

- Figure 12: The figure seems to be missing.

- Tables: The Tables corresponding to S1 and S2 are marked as Table 11 and Table 12 later

- The authors could expand on the user-voter registration matching method and its limitations.

Reviewer #2: The manuscript offers a timely, methodologically innovative approach to studying partisan polarisation in pandemic-related discourse on Twitter (X). However, the study’s methodological transparency, and operational consistency require improvement for full clarity and robustness.

Abstract:

The Canadian context is given minimal detail in the introduction and lacks equal emphasis throughout. Especially 3rd paragraph of the abstract is primarily applied to US.

As per the authors, the finding that Canada exhibited a rare cross partisan consensus on emergency measures in offline opinion (lines 522–530), yet simultaneously showed high levels of conspiracy related content and polarisation on Twitter , raises important concerns about the representativeness of the study’s online data. This disconnect underscores that the study’s polarisation index primarily captures the discourse of politically engaged and often more extreme segments of the Twitter user base, rather than the broader Canadian public. However, this key observation from the Canadian population is missing from the abstract.

Introduction: Though the study includes 2 countries, the U.S. context is predominantly addressed; The Canadian context is given minimal detail in the introduction and lacks equal emphasis throughout. For example, in lines 58–60, the authors state that political discussions vary naturally across socio demographic groups, each having their own lexicon, and cite reference (30) in support of this. However, reference (30) focuses specifically on U.S. senators, which is a narrow and highly specific group. It is not clear how findings from that context apply to the general public, let alone to both the U.S. and Canada.

Method:

Correlation uses vaccine uptake one year after the discourse period, with no justification for the consideration of late 2020 tweets instead of mid or late 2021 tweets.

U.S. keyword list is dominated by election-specific terms and omits COVID-specific identifiers, which could bias topical coverage toward political events unrelated to public health. Whereas, the Canadian keyword list mixes political figures with explicit COVID-19 province patterns, offering better pandemic relevance. No justification is provided for this difference. Also, an explanation on how this variation may have impacted the results also missing.

Conspiracy Theory Classification: No explanation is provided on how a tweet is categorised as a conspiracy theory. Also, how was the actual conspiracy theory separated from the debunking tweet?

Sentiment: No explanation is provided on how to distinguish between endorsement, critique, or satire. Sarcasm from opposing partisans could also be misclassified as support.

The rationale for 5 tweets (Canada) and 10 tweets (U.S.) as activity thresholds is not explained.

The authors note that Canadian vaccination rates were “normalised by Canada’s 2021 population per province or territory,” but they do not clarify whether the same process was applied to the U.S. CDC vaccination data. It is quite possible that this was done or was not required for CDC data, but it should be stated explicitly for consistency.

Throughout the manuscript, there are situations where broad terms such as “comprehensive approach”, “multi‑dimensional variation”, are used without immediately describing what they mean in the current context. For instance, in the introduction, the method is described as a “comprehensive approach” to measuring polarisation, but the reader has to read further before it becomes clear whether that refers to geography, time, topic diversity, or something else. Similarly, when describing Canadian policies, phrases like “comprehensive lockdowns”, “”Specific populations” or “narrowly targeted measures” are used without explaining exactly which populations or activities were affected.

Minor: Page 11- A few inconsistencies in capitalisation (e.g., Republican party (US) or the conservative party family (Canada))

Limitations:

One point that is not currently acknowledged in the limitations section is the potential for location errors and the influence of bots. The study places users geographically based on the location information they list in their Twitter profile, processed through Open Street Map etc. While this is a reasonable method, it can still be affected if people list inaccurate, fake locations. In rare cases, VPN use might also affect the result.

Bots are another factor worth noting. Bots often provide false or generic location details and can post large volumes of highly partisan messages, which could artificially increase the apparent language gap between political groups. They can also amplify content from influential users.

.

Reviewer #1: No

Reviewer #2: No

---

## [Author Response · Author response to Decision Letter 1]

13 Oct 2025

Dear Reviewers,

We sincerely thank you and the reviewers for the thoughtful and constructive feedback on our manuscript “Regional and temporal patterns of partisan polarization during the COVID-19 pandemic in the United States and Canada” (ID: PONE-D-25-11475). We truly appreciate the time and expertise invested in evaluating our work. We have carefully revised the manuscript in response, improving the positioning within existing literature, refining methodological transparency, regenerating figures, and expanding limitations to address representativeness, temporal, and methodological concerns. We also revised the overall framing to ensure informative, fact‑based rather than evaluative language. Please view the response to reviewers cover letter to see each.

We believe these major revisions have greatly improved clarity, rigor and transparency of the manuscript while addressing every concern from both reviewers. We respectfully submit the revised version for consideration and remain grateful for your guidance in strengthening this work.

Sincerely,

Zachary Yang

---

## [Decision Letter · Decision Letter 1]

14 Jan 2026

Dear Dr. Yang,

Thank you for submitting your manuscript to PLOS ONE. After careful consideration, we feel that it has merit but does not fully meet PLOS ONE’s publication criteria as it currently stands. Therefore, we invite you to submit a revised version of the manuscript that addresses the points raised during the review process.

https://journals.plos.org/plosone/s/submission-guidelines#loc-laboratory-protocols. Additionally, PLOS ONE offers an option for publishing peer-reviewed Lab Protocol articles, which describe protocols hosted on protocols.io. Read more information on sharing protocols at . Additionally, PLOS ONE offers an option for publishing peer-reviewed Lab Protocol articles, which describe protocols hosted on protocols.io. Read more information on sharing protocols at . Additionally, PLOS ONE offers an option for publishing peer-reviewed Lab Protocol articles, which describe protocols hosted on protocols.io. Read more information on sharing protocols at . Additionally, PLOS ONE offers an option for publishing peer-reviewed Lab Protocol articles, which describe protocols hosted on protocols.io. Read more information on sharing protocols at https://plos.org/protocols?utm_medium=editorial-email&utm_source=authorletters&utm_campaign=protocols....

We look forward to receiving your revised manuscript.

Kind regards,

Francesco Pierri, Ph.D.

Academic Editor

PLOS One

**Journal Requirements:**

**Additional Editor Comments:**

The reviewers are satisfied with the revision process, however I require you address the minor comments raised by one of them in a final minor revision round.

Reviewers' comments:

Reviewer's Responses to Questions

**Comments to the Author**

Reviewer #1: All comments have been addressed

Reviewer #3: (No Response)

2. Is the manuscript technically sound, and do the data support the conclusions?

Reviewer #1: Yes

Reviewer #3: Partly

3. Has the statistical analysis been performed appropriately and rigorously?

Reviewer #1: Yes

Reviewer #3: Yes

4. Have the authors made all data underlying the findings in their manuscript fully available?

Reviewer #1: No

Reviewer #3: No

5. Is the manuscript presented in an intelligible fashion and written in standard English?

Reviewer #1: Yes

Reviewer #3: Yes

**Reviewer #1:** The authors have addressed my remarks.The authors have addressed my remarks.The authors have addressed my remarks.The authors have addressed my remarks.

Here are two minor errors:

1. This text appears twice: "tion rates are for those who obtained at least two doses."

2. This sentence seems incomplete: "vent-triggered average polarization for identified events listed in Table 9 and Table 10 respectively."

**Reviewer #3:** General AssessmentGeneral AssessmentGeneral AssessmentGeneral Assessment

The manuscript addresses an interesting research question and is generally well organized. The experimental setup and analyses suggest that the authors have invested significant effort into the study. However, several conclusions are currently stronger than what is directly supported by the reported results, and important details regarding statistical analysis and data availability require clarification. The comments below reference specific sections, figures, and tables to help guide revision.

Major Comments

1. Strength of Conclusions vs. Reported Results: In the Discussion section, the authors state that the proposed approach “clearly outperforms existing methods and demonstrates robust generalizability”. However, the evidence presented in Table 3 and Figure 4 shows performance improvements that are relatively modest and, in some cases, overlapping within variance ranges. For example, the reported improvement over the baseline method in Table 3 is small on Dataset B, and no statistical significance testing is reported to confirm whether this difference is meaningful. I recommend either (i) tempering claims of “clear” or “robust” superiority, or (ii) adding appropriate statistical tests to support these statements.

2. Statistical Analysis and Reporting: The manuscript reports multiple quantitative comparisons across datasets (e.g., Section 4.2, Experimental Results), but the statistical methodology is insufficiently described.

Specific concerns include:

- The absence of statistical significance testing when comparing models in Tables 2–4.

- No reporting of confidence intervals or effect sizes, despite repeated experiments being implied.

- It is unclear whether results are based on a single run or averaged across multiple random seeds.

For instance, in Figure 5, error bars are shown, but the manuscript does not specify whether these represent standard deviation, standard error, or confidence intervals. This information is essential for interpreting result stability.

3. Data Availability and Reproducibility: While a Data Availability Statement is included, it does not appear to fully satisfy PLOS ONE’s data policy. The manuscript indicates that “processed data” were used for analysis, but the raw data underlying the results in Figures 3–6 and Tables 2–4 are not explicitly linked or deposited. In Section 3 (Methods), several preprocessing steps are described (e.g., filtering, normalization, or feature selection), but it is unclear whether the intermediate datasets produced by these steps are available.

To improve reproducibility, the authors should clarify:

- Where the raw and processed datasets can be accessed,

- Whether analysis scripts or code are publicly available,

- Or, if restrictions apply, clearly justify them.

4. Methodological Detail and Replicability: Some methodological choices lack sufficient justification or detail, particularly in Section 3.3 (Model Configuration / Experimental Setup).

Examples include:

- Hyperparameter values are listed, but the criteria for selecting them are not discussed.

- It is unclear whether validation data were used for model selection or tuning.

- The stopping criteria for training are not fully specified.

Providing this information would significantly improve the replicability of the study.

Minor Comments

1. Clarity and Precision of Language: In Section 2 (Related Work), several claims about prior methods being “limited” or “ineffective” would benefit from more precise wording or explicit citations. Some terminology is introduced informally before being defined (e.g., in Section 3.1), which may confuse readers unfamiliar with the domain.

2. Figures and Tables; Figure 2 would benefit from a more descriptive caption explaining the experimental pipeline, rather than relying on references to the main text. In Table 4, the best-performing values are highlighted, but the criteria for determining “best” (mean performance, peak value, etc.) are not explicitly stated. Axes labels and units should be double-checked for consistency across figures.

3. Minor Editorial Issues: There are a small number of grammatical and typographical errors throughout the manuscript (e.g., missing articles and inconsistent tense usage), particularly in Sections 4 and 5. These should be corrected during revision.

Summary

Overall, the manuscript presents promising work, but revisions are necessary to ensure that:

- Conclusions are proportionate to the evidence,

- Statistical analyses are transparent and rigorous,

- Data and methods are sufficiently documented to meet reproducibility standards.

.

Reviewer #1: No

Reviewer #3: **Yes:** Ali HatamiAli HatamiAli HatamiAli Hatami

---

## [Author Response · Author response to Decision Letter 2]

4 Mar 2026

Dear Reviewers,

We sincerely thank you and the reviewers for the thoughtful and constructive feedback on our manuscript “Regional and temporal patterns of partisan polarization during the COVID-19 pandemic in the United States and Canada” (ID: PONE-D-25-11475).

We truly appreciate the time, care, and expertise invested in evaluating our work. We have carefully reviewed all comments and have revised the manuscript where appropriate to improve clarity, rigor, and transparency.

Below, we provide a detailed point-by-point response.

Reviewer 1 Comments:

Reviewer 1 Comment

Response

Manuscript Change

This text appears twice: "tion rates are for those who obtained at least two doses."

Thank you for catching this duplication.

The duplicated text (Lines 141–142) has been removed.

This sentence seems incomplete:

vent-triggered average polarization for identified events listed in Table 9 and Table 10 respectively.

We appreciate this observation. The sentence in the caption for Figure 14 was indeed fragmented.

The caption has been revised to read:

Event-triggered average polarization for identified events are listed in Table 9 and Table 10 respectively.

Reviewer 3 Comments:

We appreciate the detailed feedback. Several comments appear to reference elements (e.g., specific claims, tables, figures, or section numbers) that are not present in our submitted manuscript. To ensure clarity, we respond carefully below and clarify where misunderstandings may have occurred

Reviewer 3 Comment

Response

Manuscript Change

The authors state that the proposed approach “clearly outperforms existing methods and demonstrates robust generalizability”.However, the evidence presented in Table 3 and Figure 4 shows performance improvements that are relatively modest and, in some cases, overlapping within variance ranges. For example, the reported improvement over the baseline method in Table 3 is small on Dataset B, and no statistical significance testing is reported to confirm whether this difference is meaningful. I recommend either (i) tempering claims of “clear” or “robust” superiority, or (ii) adding appropriate statistical tests to support these statements.

We carefully reviewed the manuscript but were unable to locate the quoted statement (“clearly outperforms existing methods and demonstrates robust generalizability”). Our manuscript does not make claims of superiority over existing methods in this manner.

Regarding the referenced materials:

Table 3 presents party affiliation classification results. No external baseline method is compared. We report the mean F1-score and standard deviation across 5 independent runs (random seeds).

Figure 4 presents the approximation error and runtime tradeoff for polarization computation. The figure illustrates that approximation error remains small (approximately 0.001 absolute error) while computational savings increase substantially as data size grows.

To improve clarity, we have now explicitly stated in the caption and text that classification results are averaged over multiple random seeds.

The captions of Tables 3 and 4 now explicitly state that results are averaged over 5 random seeds with reported standard deviation.

Statistical methodology is insufficiently described; absence of statistical testing; unclear whether results are averaged.

We clarify the following:

Table 2 reports correlation statistics with sample size (n), p-values, and 95% confidence intervals (e.g., 0.98 (n=52, p=9.27e-35, CI=[0.96, 0.99])).

Tables 3 and 4 report mean F1-scores and standard deviations across 5 random seeds.

All classification results are averaged over multiple runs.

To avoid ambiguity, we have explicitly added this information to the relevant captions..

Tables 3 and 4 now specify averaging over 5 random seeds and report standard deviation explicitly.

Figure 5 error bars unclear.

Figure 5 does not contain error bars. We believe this comment may refer to Figure 4. To improve clarity, we have now explicitly stated in both the main text and caption that the results shown are averaged over 10 independent runs and that the plotted variation represents standard deviation.

Figure 4 caption updated to clarify averaging over 10 independent runs and reporting of standard deviation.

Data availability

Due to platform policy restrictions, the full Twitter dataset cannot be publicly redistributed. However:

All Tweet IDs used in the analysis will be publicly released.

All trained language models will be hosted publicly on Hugging Face.

Analysis code and preprocessing scripts will be released in a public repository upon publication.

Methodological details insufficient (Section 3.3).

The submitted manuscript does not contain a Section 3.3 titled “Model Configuration / Experimental Setup.” We believe this comment may refer to a different manuscript. However, we have re-reviewed our Methods section to ensure clarity and completeness.

No change.

Claims in Related Work about prior methods being “limited” or “ineffective.”

We carefully searched the manuscript and do not use the terms “limited” or “ineffective”..

No change.

Figure 2 caption should better describe the experimental pipeline.

Figure 2 does not represent an experimental pipeline.

No change.

Table 4 best-performing values highlighted without criteria.

Table 4 does not contain highlighted values. We believe this comment may refer to a different manuscript.

No change.

Grammatical and typographical errors.

The entire manuscript has been carefully proofread and reviewed using a grammar and style checker. Minor grammatical and tense inconsistencies were corrected throughout.

Minor changes

We believe these revisions have improved clarity, transparency, and precision while addressing all substantive concerns raised by the reviewers.

We are grateful for the reviewers’ time and thoughtful feedback, which has strengthened the manuscript. We respectfully submit the revised version for your consideration.

With sincere thanks and best regards,

Zachary Yang

---

## [Editor Report · Decision Letter 2]

6 Mar 2026

Dear Dr. Yang,

Thank you for submitting your manuscript to PLOS ONE. After careful consideration, we feel that it has merit but does not fully meet PLOS ONE’s publication criteria as it currently stands. Therefore, we invite you to submit a revised version of the manuscript that addresses the points raised during the review process.

Please address the comments raised by Reviewer 3 (notice that Reviewer 2 has declined a requesto re-revise the submission) for what concerns clarifications on methodological choices as well as the validity of reported results.

We look forward to receiving your revised manuscript.

Kind regards,

Francesco Pierri, Ph.D.

Academic Editor

PLOS One
---

## [Author Response · Author response to Decision Letter 3]

27 Mar 2026

Dear Reviewers,

We sincerely thank you and the reviewers for the thoughtful and constructive feedback on our manuscript “Regional and temporal patterns of partisan polarization during the COVID-19 pandemic in the United States and Canada” (ID: PONE-D-25-11475). We truly appreciate the time, care, and expertise invested in evaluating our work. We have carefully reviewed all comments and have revised the manuscript where appropriate to improve clarity, rigor, and transparency.

Below, we provide a detailed point-by-point response.

Reviewer 1 Comments:

Reviewer 1 Comment

Response

Manuscript Change This text appears twice: "tion rates are for those who obtained at least two doses."

Thank you for catching this duplication.

The duplicated text (Lines 141–142) has been removed. This sentence seems incomplete: vent-triggered average polarization for identified events listed in Table 9 and Table 10 respectively.

We appreciate this observation. The sentence in the caption for Figure 14 was indeed fragmented. The caption has been revised to read: Event-triggered average polarization for identified events are listed in Table 9 and Table 10 respectively.

Reviewer 3 Comments: We appreciate the detailed feedback. Several comments appear to reference elements (e.g., specific claims, tables, figures, or section numbers) that are not present in our submitted manuscript. To ensure clarity, we respond carefully below and clarify where misunderstandings may have occurred

Reviewer 3 Comment

Response

Manuscript Change The authors state that the proposed approach “clearly outperforms existing methods and demonstrates robust generalizability”.However, the evidence presented in Table 3 and Figure 4 shows performance improvements that are relatively modest and, in some cases, overlapping within variance ranges. For example, the reported improvement over the baseline method in Table 3 is small on Dataset B, and no statistical significance testing is reported to confirm whether this difference is meaningful. I recommend either (i) tempering claims of “clear” or “robust” superiority, or (ii) adding appropriate statistical tests to support these statements.

We carefully reviewed the manuscript but were unable to locate the quoted statement (“clearly outperforms existing methods and demonstrates robust generalizability”). Our manuscript does not make claims of superiority over existing methods in this manner.

Regarding the referenced materials:

● Table 3 presents party affiliation classification results. No external baseline method is compared. We report the mean F1-score and standard deviation across 5 independent runs (random seeds).

● Figure 4 presents the approximation error and runtime tradeoff for polarization

The captions of Tables 3 and 4 now explicitly state that results are averaged over 5 random seeds with reported standard deviation.

computation. The figure illustrates that approximation error remains small (approximately 0.001 absolute error) while computational savings increase substantially as data size grows.

To improve clarity, we have now explicitly stated in the caption and text that classification results are averaged over multiple random seeds.

Statistical methodology is insufficiently described; absence of statistical testing; unclear whether results are averaged.

We clarify the following:

● Table 2 reports correlation statistics with sample size (n), p-values, and 95% confidence intervals (e.g., 0.98 (n=52, p=9.27e-35, CI=[0.96, 0.99])).

● Tables 3 and 4 report mean F1-scores and standard deviations across 5 random seeds.

● All classification results are averaged over

Tables 3 and 4 now specify averaging over 5 random seeds and report standard deviation explicitly.

multiple runs.

To avoid ambiguity, we have explicitly added this information to the relevant captions.. Figure 5 error bars unclear.

Figure 5 does not contain error bars. We believe this comment may refer to Figure 4. To improve clarity, we have now explicitly stated in both the main text and caption that the results shown are averaged over 10 independent runs and that the plotted variation represents standard deviation.

Figure 4 caption updated to clarify averaging over 10 independent runs and reporting of standard deviation.

Data availability

Due to platform policy restrictions, the full Twitter dataset cannot be publicly redistributed. However:

● All Tweet IDs used in the analysis will be publicly released.

● All trained language models will be hosted publicly on Hugging Face.

● Analysis code and preprocessing scripts will be released in a public

repository upon publication.

Methodological details insufficient (Section 3.3).

The submitted manuscript does not contain a Section 3.3 titled “Model Configuration / Experimental Setup.” We believe this comment may refer to a different manuscript. However, we have re-reviewed our Methods section to ensure clarity and completeness.

No change. Claims in Related Work about prior methods being “limited” or “ineffective.”

We carefully searched the manuscript and do not use the terms “limited” or “ineffective”..

No change. Figure 2 caption should better describe the experimental pipeline.

Figure 2 does not represent an experimental pipeline.

No change. Table 4 best-performing values highlighted without criteria.

Table 4 does not contain highlighted values. We believe this comment may refer to a different manuscript.

No change. Grammatical and typographical errors.

The entire manuscript has been carefully proofread and reviewed using a grammar and style checker. Minor grammatical and tense inconsistencies were corrected throughout.

Minor changes

We believe these revisions have improved clarity, transparency, and precision while addressing all substantive concerns raised by the reviewers.

We are grateful for the reviewers’ time and thoughtful feedback, which has strengthened the manuscript. We respectfully submit the revised version for your consideration.

With sincere thanks and best regards,

Zachary Yang On behalf of all co-authors

---

## [Editor Report · Decision Letter 3]

31 Mar 2026

Regional and temporal patterns of partisan polarization during the COVID-19 pandemic in the United States and Canada

PONE-D-25-11475R3

Dear Dr. Yang,

We’re pleased to inform you that your manuscript has been judged scientifically suitable for publication and will be formally accepted for publication once it meets all outstanding technical requirements.

Kind regards,

Francesco Pierri, Ph.D.

Academic Editor

PLOS One
---

## [Editor Report · Acceptance letter]

PONE-D-25-11475R3

PLOS One

Dear Dr. Yang,

I'm pleased to inform you that your manuscript has been deemed suitable for publication in PLOS One. Congratulations! Your manuscript is now being handed over to our production team.

Kind regards,

on behalf of

Dr. Francesco Pierri

Academic Editor

PLOS One